# Multi-level, forming and filament free, bulk switching trilayer RRAM for neuromorphic computing at the edge

Jaeseoung Park [1,6], Ashwani Kumar[1,6], Yucheng Zhou[1], Sangheon Oh[1], Jeong-Hoon Kim[1], Yuhan Shi[1], Soumil Jain[2], Gopabandhu Hota[1], Erbin Qiu[3], Amelie L. Nagle[4], Ivan K. Schuller[3], Catherine D. Schuman [5], Gert Cauwenberghs[2] & Duygu Kuzum [1] ✉

CMOS-RRAM integration holds great promise for low energy and high throughput neuromorphic computing. However, most RRAM technologies relying on filamentary switching suffer from variations and noise, leading to computational accuracy loss, increased energy consumption, and overhead by expensive program and verify schemes. We developed a filament-free, bulk switching RRAM technology to address these challenges. We systematically engineered a trilayer metal-oxide stack and investigated the switching characteristics of RRAM with varying thicknesses and oxygen vacancy distributions to achieve reliable bulk switching without any filament formation. We demonstrated bulk switching at megaohm regime with high current nonlinearity, up to 100 levels without compliance current. We developed a neuromorphic compute-in-memory platform and showcased edge computing by implementing a spiking neural network for an autonomous navigation/racing task. Our work addresses challenges posed by existing RRAM technologies and paves the way for neuromorphic computing at the edge under strict size, weight, and power constraints.

As the Moore's law is coming to an end due to the limitations of physical scaling of CMOS technology, neuromorphic compute-in-memory (CIM) approaches have attracted huge attention to keep improving computing performance[1]. The CIM has the potential to alleviate the von Neumann bottleneck, a limitation in computing performance resulting from significant energy loss and time delays during data transfer between processors and memory units in classical computing systems. While GPUs and tensor processing units excel in parallel computing compared to CPUs, they are still reliant on static random access memory, which demands substantial physical space[2,3]. Emerging non-volatile memory (eNVM) devices including phase change memory (PCM)[4], magnetic random access memory (MRAM)[5],

conductive bridge random access memory (CBRAM)[6,7], ferroelectric field effect transistor (FeFET)[8], resistive random access memory (RRAM)[9,10], and memristive synapses based on 2D materials[11–13] have been extensively studied for physical implementations of neuromorphic CIM platforms. RRAM devices are gaining attention due to their exceptional density, lower fabrication cost, and back-end-of-line (BEOL) compatibility with CMOS technology[9,14].

RRAM-based reconfigurable systems hold great promise for low energy and high throughput neuromorphic computing. Neurosynaptic cores constructed by CMOS-RRAM integration have shown dynamically high-performance reconfigurable dataflow and energy efficiency of 74 TMACS/W[15,16]. However, three major challenges are yet to be addressed

[1]Department of Electrical and Computer Engineering, University of California San Diego, La Jolla, CA, USA. [2]Department of Bioengineering, University of California San Diego, La Jolla, CA, USA. [3]Department of Physics, University of California San Diego, La Jolla, CA, USA. [4]Department of Computer Science, Massachusetts Institute of Technology, Cambridge, MA, USA. [5]Department of Electrical Engineering and Computer Science, University of Tennessee, Knoxville, TN, USA. [6]These authors contributed equally: Jaeseoung Park, Ashwani Kumar. ✉e-mail: dkuzum@ucsd.edu

to scale CMOS-RRAM based accelerators and achieve energy-efficient dynamic on-chip learning with RRAM crossbar arrays: (i) Most of the RRAM devices rely on filamentary switching, which suffers from extensive variations and noise leading to computational accuracy loss and increased energy consumption[17]. Programming RRAM into multi-level resistance states requires expensive read and verify programming schemes, unsuitable for on-chip training[18,19]. (ii) Low ON-state resistance of filamentary RRAM increases the power consumption due to high current read and write operations. As the resistance approaches the interconnect resistance[20], it constrains the array size and parallel multiply & accumulate (MAC) operations. (iii) Filamentary RRAM requires high forming voltages to generate a conductive filament, that is not compatible with advanced CMOS technology nodes. To address all these challenges, here, we demonstrate systematic engineering of a trilayer metal-oxide bulk RRAM stack and investigate the switching characteristics of RRAM devices with varying thicknesses and $V_O$ distributions across the trilayer. Sputtered porous $TiO_x$ layer facilitates modulation of $V_O$ distribution in the switching layer without forming $V_O$ filaments (Fig. 1a), enabling bulk switching operations in the megaohm (M$\Omega$) range, achieving high current nonlinearity, and programming up to 100 levels without the need for compliance current. Highly linear MVMs are achieved by using the row-differential scheme instead of non-differential scheme in fabricated bulk RRAM crossbars[21]. We employ the fabricated RRAM crossbars to perform control for an autonomous navigation/racing task using a spiking neural network (SNN) model, demonstrating compatibility for neuromorphic computing at the edge applications. Our work tackles the challenges presented by current filamentary RRAM technologies, clearing a path for neuromorphic computing at the edge while adhering to stringent size, weight, and power constraints.

## Results

### Optimization of trilayer bulk RRAM stack
To systematically investigate switching characteristics of RRAM devices based on multi-layer stacks, we fabricated RRAM devices in four different switching layer ($Al_2O_3$/$TiO_2$/$TiO_x$) configurations (Table 1). Our detailed fabrication process is explained in the methods. All samples include 3 nm $Al_2O_3$ as a high bandgap tunnel barrier ($E_g$ ~9.0 eV) layer to limit the current and provide I-V nonlinearity through tunneling. For S1 and S2, ALD $TiO_2$ layers (S1 = 20 nm, S2 = 40 nm) were deposited without breaking the vacuum. S3 and S4 have 3 nm ALD $TiO_2$ and sputtered $TiO_x$ layers (S3 = 6.5 nm, S4 = 40 nm) with varying oxygen stoichiometry (Fig. 1a). As shown in the plane-view and cross-sectional SEM images, 16×16 crossbar arrays were fabricated using a via-hole structure design (Fig. 1b–d). The via-hole structure was chosen to achieve uniform and reliable device switching instead of simple crossbar structure. The via-hole design with 150 nm thick plasma-enhanced chemical vapor deposition (PECVD) $SiO_2$ insulator eliminates the edge effects due to high-field corners or sidewalls[22]. In addition, all steps of the fabrication process have low thermal budget (T < 300 °C) which is perfectly compatible with CMOS BEOL integration process.

We first tested DC switching characteristics for all samples. Both S1 (Fig. 1e, $R_{on}$ = 150 $\Omega$, $R_{off}$ = 400 k$\Omega$) and S2 (Fig. 1f, $R_{on}$ = 3 k$\Omega$, $R_{off}$ = 2 G$\Omega$) exhibit only filamentary switching with significant variations in set/reset voltages in consistent with the previous research on the filamentary RRAM using $Al_2O_3$/$TiO_{2-x}$ stacks[9]. The high OFF-state resistance of S2 is due to thicker $TiO_2$ layer. A 200$\mu$A compliance current is necessary to prevent permanent breakdown during DC set process for both devices. Although the filamentary RRAM shows resistance switching behavior, these devices suffer from highly non-uniform switching characteristics due to the stochastic nature of filament formation and rupture[17]. The low ON-state resistance of the filamentary RRAM also increases the power consumption due to high energy read and write operations. In addition, the abrupt resistance jumps during the set and reset processes are not suitable for continuous synaptic weight updates during online learning where the multi-level conductance update is needed. RRAM devices including sputtered $TiO_2$ (S3) or $TiO_x$ (S4) layer exhibit bulk switching characteristics (Fig. 1g, h). Surprisingly, S3 shows both filamentary and bulk

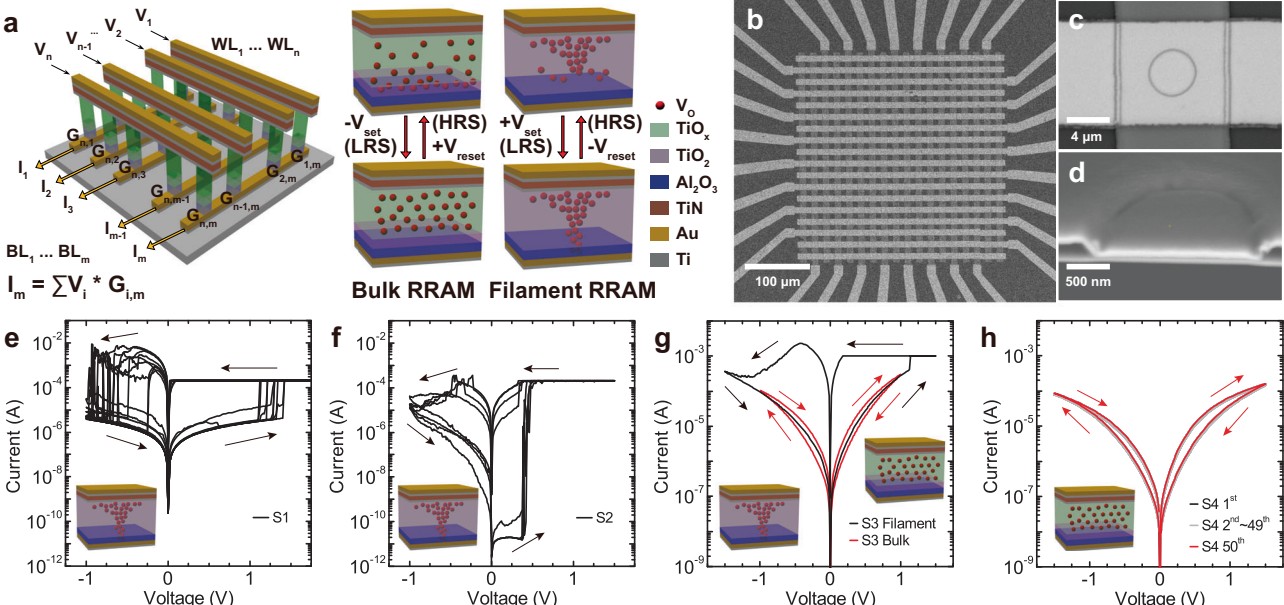

**Fig. 1 | RRAM device stack and DC I-V switching characterization. a** Illustration of fabricated RRAM device stack and crossbar arrays. Bulk and filamentary RRAM switching mechanisms are compared. For bulk switching, the distribution of oxygen vacancies ($V_O$) is modulated between $TiO_x$ and $TiO_2$ layers. For filamentary switching, the $V_O$ filament formation and rupture occur near the bottom electrode. Scanning Electron Microscopy (SEM) images of fabricated **b** 16×16 crossbar array **c** single trilayer RRAM device, and **d** cross-section of a half-cut RRAM device. Filamentary switching characteristics of **e** S1 and **f** S2. **g** Coexistence of filamentary and bulk switching RRAM in S3. They show the opposite polarity due to the different resistance-switching mechanisms. Black arrows show the polarity of filamentary switching, while red arrows show polarity of bulk switching. **h** Bulk RRAM DC I-V characteristics of S4 without forming filaments. 50 cycles of DC sweeps perfectly overlap, showing highly uniform bulk switching.

switching with a transition from bulk switching to filamentary switching as DC sweep range is increased from 1 V to 1.5 V. In high voltage DC sweep range ($|V| < 1.5\,V$) where filamentary switching is observed, it follows the same switching polarity (a positive set and negative reset voltage) as S1 and S2 filamentary RRAM devices ($R_{on} = 200\,\Omega$, $R_{off} = 145\,k\Omega$). In low voltage DC sweep range ($|V| < 1\,V$) where bulk switching dominates, it demonstrates gradual resistance change during DC sweep without any sudden resistance jumps that are observed in filamentary switching ($R_{on} = 76\,k\Omega$, $R_{off} = 180\,k\Omega$). Switching direction for bulk switching (a negative voltage set and a positive voltage reset) show opposite polarity to filamentary switching. Although both filamentary and bulk switching are observed in different voltage regimes and the opposite polarity, coexistence of both mechanisms is not desirable for reliable synaptic weight updates[23]. For the RRAM devices with a thicker and $V_O$-rich sputtered $TiO_x$ layer (S4), only bulk switching behavior is observed without any filament formation. S4 exhibits, highly reliable bulk switching behavior with excellent uniformity over 50 DC cycles (Fig. 1h, $R_{on} = 410\,k\Omega$, $R_{off} = 1\,M\Omega$). Furthermore, the bulk switching for S4 exists in the $M\Omega$ resistance range in contrast to bulk switching occurring ~100 $k\Omega$ for S3. Therefore, we decided to further investigate switching characteristics and multi-level resistance states for the trilayer bulk RRAM (S4) and chose it for neuromorphic computing with the crossbar array demonstrations.

To analyze trilayer structure of the bulk switching RRAM, the transmission electron microscopy (TEM) and scanning transmission electron microscopy - electron energy loss spectroscopy (STEM-EELS) analyses were performed (Fig. 2). We estimated the composition of Ti

metal layer as $TiO_{1.2}$ based on the composition analysis. The top Ti metal layer scavenges the oxygen from the sputtered $TiO_x$ layer due to the lower chemical potential of oxygen in Ti suboxides than that in $TiO_2$[24]. There are previous studies that exploit Ti as a scavenging layer to reduce underlying oxide layers[25,26]. For example, for the Nb-based selector device fabrication, Ti metal plays an important role in stabilizing the underlying $NbO_2$ selector layer without further oxidation to the thermally stable $Nb_2O_5$ composition. The Ti metal also reduces Hf-based oxides to induce oxygen vacancy defects in it so that the RRAM device can form the filaments at the lower set voltage. The ALD $TiO_2$ layer has darker contrast in bright field-TEM image, confirming higher atomic density than the $TiO_x$ layer (Fig. 2a). STEM-EELS line-scan profile shows lower oxygen concentration in $TiO_x$ layer than ALD $TiO_2$ layer (Fig. 2b). Furthermore, STEM-EELS composition map (Fig. 2c) shows nm-scale dark areas only in the sputtered $TiO_x$ layer pointing to a porous structure. To further analyze the crystal structure and film density, grazing incidence X-ray diffraction (GIXRD) and X-ray reflection (XRR) measurements were conducted (Supplementary Fig. S1). 30 nm ALD $TiO_2$ layer shows crystalline anatase phase while sputtered $TiO_x$ films show an amorphous phase. The grain boundaries are well known to be the high diffusivity paths of small ions such as oxygens or hydrogens[27,28]. Especially in polycrystalline filament RRAM devices, the grain boundaries acts an important role in charge transport and $V_O$ accumulation and diffusion[29]. Due to these diffusivity paths, filament formation and rupture easily occur in the filamentary RRAM devices (S1 and S2). In the amorphous phase, however, there are no high diffusivity paths for $V_O$, so the filament formation can be successfully suppressed. XRR measurements show that the critical angle of sputtered $TiO_x$ layer (0.52°) is smaller than ALD $TiO_2$ layer (0.55°), suggesting that the film mass density is smaller for the sputtered $TiO_x$ layer. The distribution of $V_O$ defects is modulated by the external electric field in a whole switching layer rather than forming a locally accumulated $V_O$ filaments, so that we can achieve bulk switching behavior.

### Characterization of bulk RRAM switching behavior

Observing uniform and forming free bulk switching in the trilayer RRAM with oxygen deficient $TiO_x$ layer, we investigated the area scaling of the device resistance to confirm bulk switching (Fig. 3a–d, Diameter: 3–10 μm). For the trilayer RRAM (S4), resistance linearly

**Table 1 | Four different multilayer stacks are fabricated (S1–S4). Only the trilayer with a sputtered $TiO_x$ layer (S4) shows stable bulk switching characteristics without filament formation**

| Sample | Oxide stack | Dominating switching |
|---|---|---|
| S1 | ALD $Al_2O_3$/$TiO_2$ (3 nm/20 nm) | Filamentary |
| S2 | ALD $Al_2O_3$/$TiO_2$ (3 nm/40 nm) | Filamentary |
| S3 | ALD $Al_2O_3$/$TiO_2$ (3 nm/3 nm) / Sputter $TiO_x$ (6.5 nm) | Filamentary/Bulk |
| S4 | ALD $Al_2O_3$/$TiO_2$ (3 nm/3 nm) / Sputter $TiO_x$ (40 nm) | Bulk |

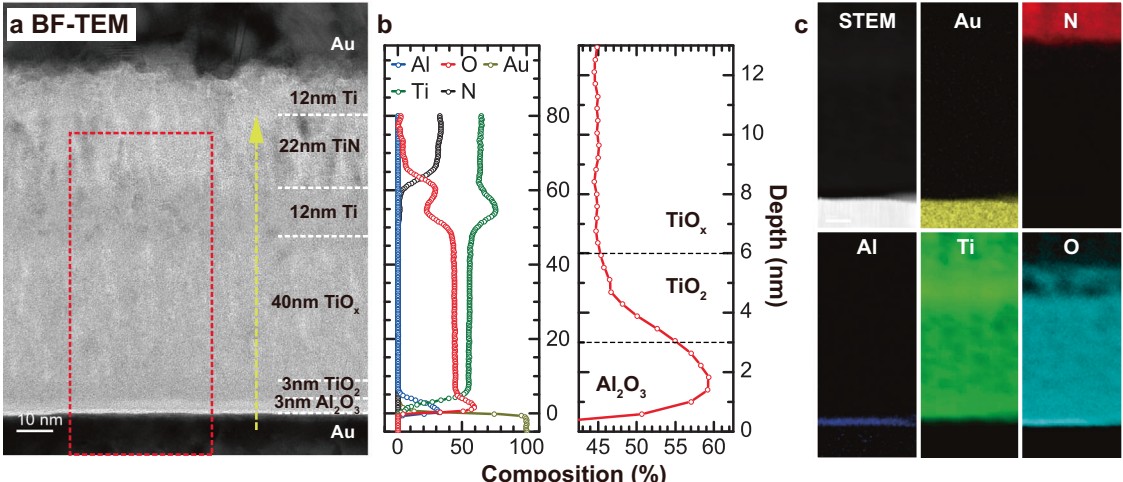

**Fig. 2 | Cross-sectional analysis of bulk RRAM device. a** Cross-sectional bright-field Transmission Electron Microscopy (TEM) image of trilayer bulk RRAM. The bright contrast of $TiO_x$ suggests a porous structure for the layer. **b** Atomic concentration profile measured by Scanning Transmission Electron Microscopy – Electron Energy Loss Spectroscopy (STEM-EELS) along the yellow arrow. All interfaces were determined based on the ion concentration and contrast in TEM image. The sputtered $TiO_x$ layer shows a smaller oxygen concentration which is lower than ALD $TiO_2$ layer due to the $V_O$ in the layer. **c** STEM-EELS mapping of red dotted box region in **a**.

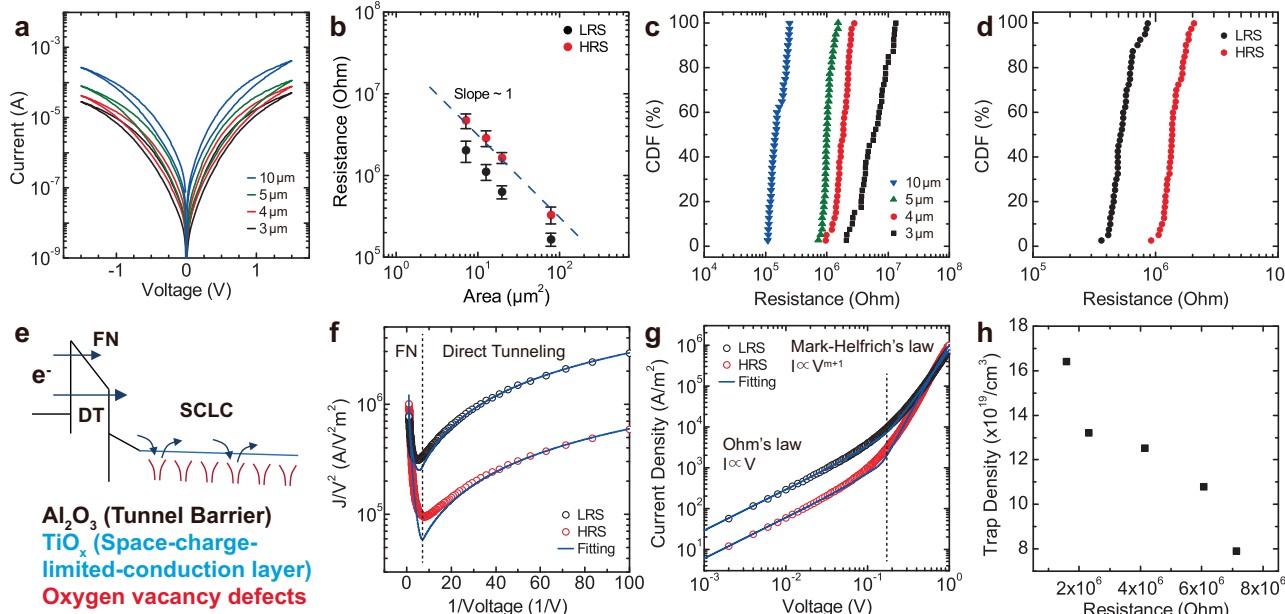

**Fig. 3 | Electrical DC characterization of bulk RRAM devices. a** DC I-V switching curves of trilayer bulk RRAM with different diameter cells from 3 to 10 μm. **b** Double log plot of resistance vs. cell area. Area-scaling behavior with a slope of 1 suggests bulk switching of RRAM devices. Each size cell data is collected from 40 different devices measured at $V_{read} = 0.1$ V. **c** Cumulative distribution function (CDF) of bulk RRAM pristine resistance in different size cells. **d** CDF of LRS and HRS states programmed with DC sweep using bulk RRAM 5μm devices. **e** Band diagram of trilayer bulk RRAM. $Al_2O_3$ 3 nm wide gap layer acts as a tunneling barrier where

the direct/Fowler-Nordheim (FN) tunneling happen at small/large voltage region. In the $TiO_x$ layer, space-charge-limited-conduction (SCLC) occurs due to deep-level $V_O$ defects. **f** Log($J/V^2$) vs. 1/V curves of high resistance state (HRS) and low resistance state (LRS). Both states show similar conduction mechanisms. **g** Log J – log V plot of LRS and HRS states. In low voltage regime, current density follows the Ohm's law ($J\propto V$), while it follows the Mark-Helfrich's law ($J\propto V^{m+1}$) in high voltage regime. **h** Trap density ($N_t$) vs. device resistance curve. $N_t$ is achieved by the fitting the experimentally measured data with our electrical conduction model.

scales with the area for both high resistance (HRS) and low resistance states (LRS) (Fig. 3b), suggesting bulk switching[23,30]. In the filamentary RRAM, both HRS and LRS resistances are independent on the device area because the resistance depends on the width and conductivity of the filament that can only be modulated by the compliance current during the SET process[31]. The device-to-device (D2D) variations of pristine, HRS, and LRS states show tight distributions in MΩ regime (Fig. 3c, d), addressing the high variability issue of filamentary RRAM devices. Perfectly overlapping DC sweeps over 50 cycles (Fig. 1h) suggest that the trilayer bulk RRAM exhibits minimal cycle-to-cycle variation.

We systematically studied the conduction mechanism of trilayer bulk RRAM by fitting DC I-V characteristics with direct tunneling, Fowler-Nordheim (FN) tunneling and space-charge-limited conduction (SCLC) models in both HRS and LRS (Fig. 3e–h). To investigate the conduction mechanism in the bulk RRAM devices, the I-V characteristics are plotted in a log($I/V^2$) vs. 1/V form (Fig. 3f). In this plot, there are two different voltage regime where the direct tunneling and Fowler-Nordheim (FN) tunneling dominate by the following relations (Direct tunneling, $I\propto V$/FN tunneling, $I\propto V^2 \cdot exp(-1/V)$)[32,33]. We developed a model explaining current conduction in our bulk RRAM devices (Supplementary note 1) and fitted to experimental results shown in Fig. 3f, g. To further validate our model, we performed temperature-dependent I-V measurements on our bulk RRAM devices and fitted the measurement results to our model (Supplementary Fig. S2). Our model based on direct tunneling, Fowler-Nordheim tunneling, and SCLC shows great agreement with the current voltage characteristics at all temperatures and voltage ranges.

In the low voltage regime (V < 0.06 V), the current of both HRS and LRS states are linear to the voltage, meaning that the direct tunneling is the dominant conduction mechanism. In the high voltage regime (V > 0.5 V), however, the log ($I/V^2$) is linear to the 1/V, where the FN tunneling becomes dominant. These tunneling conductions occur

through the high band gap $Al_2O_3$ layer which provides the MΩ-level resistance switching and high nonlinearity ($I_V/I_{0.5V} = 15$ (V = 1.5 V)) of I-V curves[34]. Both HRS and LRS states of bulk RRAM devices follow the same conduction mechanism, whereas the filamentary RRAM or CBRAM would show ohmic conduction in LRS states because current conduction occurs through the $V_O$ or metal cation filaments[35,36]. To study the switching mechanism, the double-log plot of I-V curves were fitted with the SCLC theory[37–39] (Fig. 3g). In the low voltage regime, the double-log I-V curves follows a linear relationship due to the dominance of the electron drift across $TiO_x$ layer ($I \propto V$). In the high voltage regime, they follow a power dependency on voltage due to the trap-limited conduction through the $V_O$ deep defects in the sputtered $TiO_x$ layer ($I \propto V^{m+1}$, m = $T_c$/T, $T_c$ is the characteristic temperature). Experimental I-V measurement data were fitted using our model (Supplementary note 1), and the trap density ($N_t$) was extracted. Figure 3h shows that the trap density is decreased as the device resistance is increased.

Bulk switching can be better understood by reviewing filamentary RRAM first. In the filamentary RRAM, $V_O$ defects are well known to be mobile with external electrical and thermal stimuli[40]. Filamentary RRAM needs an initial electroforming step which forms the $V_O$ defect filaments between two electrodes. Once the filaments are formed bipolar switching takes place due to the forming and rupturing of the filaments. Meanwhile, the grain boundaries are well known to be the high diffusivity paths of small ions such as oxygens or hydrogens in crystalline oxides[27,28]. Especially for our polycrystalline filamentary RRAM devices (S1, S2), the grain boundaries play an important role in charge transport and $V_O$ accumulation and diffusion. Due to these diffusivity paths, filament formation and rupture easily occur in the filamentary RRAM devices.

In our bulk RRAM devices, we deposited an amorphous, porous, and $V_O$-rich thick $TiO_x$ layer instead of having a crystalline ALD $TiO_2$ layer. Due to the absence of fast diffusion paths or accumulation

sites for $V_O$ defects in the amorphous phase, the $V_O$ defects are not clustered in specific locations that facilitate filament formation. The filament formation is effectively suppressed in an amorphous layer as compared to the crystalline phase in RRAM devices[41]. $V_O$ defects will drift homogeneously throughout the entire area of the layer rather than forming defects-clustered filaments, following the direction of the electric field, enabling bulk switching instead of filamentary switching. When a positive voltage is applied to the top electrode, the $V_O$ are pushed downwards towards the bottom electrode and the $V_O$ concentration in the $TiO_x$ layer is reduced. Space-charge-limited-conduction (SCLC) dominates the conduction in the $TiO_x$ layer. Since the $V_O$ concentration in the $TiO_x$ layer is reduced, the SCLC current decreases confirming that the device is reset to a higher resistance state. Our fitting results shown in Fig. 3g and reduced trap density shown in Fig. 3h confirms this model for the bulk switching mechanism. The bulk switching shows the opposite polarity to the filamentary switching consistent with other previous reports[23,42].

The ability to perform analog weight updates is a crucial feature in synaptic devices for efficient implementation of learning and inference in neuromorphic computing applications. Analog weight update is the most important property in synaptic devices to achieve successful neuromorphic computing applications. The filamentary RRAM shows abrupt resistance change so that they have been mainly employed for binary or low-precision implementation of neural network weights. Programming filamentary RRAM devices into discrete conductance states require extensive number of program and verify operations, not suitable for online learning applications[30]. In contrast, for the bulk RRAM devices, it is easier to achieve gradual weight updates. We first investigated gradual weight updates using identical pulses in two different conductance regimes; ~0.8 µS and ~0.12 µS. 32-states are achieved by applying of identical set and reset pulses for both conductance regimes (Fig. 4a, b). The long-term potentiation (LTP) and the long-term depression (LTD) curves show gradual conductance change ($V_{read} = 0.1$ V). We also implemented an incremental pulse scheme. We optimized the incremental pulse scheme to have linear LTP and LTD curves with a higher dynamic range and larger number of states (Fig. 4c). Figure 4d, e shows the gradual current increase/decrease during the transient set ($-2.0$ V)/reset ($+1.5$ V) pulses. We quantitatively analyzed the device non-linearity and found that incremental pulse scheme can improve non-linearity (Supplementary Fig. S3, Supplementary note 2). The non-linearity could be improved by further optimizing the pulse amplitude and width for potentiation and depression. To compensate for the non-linearity effect in hardware implementation of neural networks, we previously developed the adaptive quantization method, which maps neural network weights onto the device conductances based on the distribution and relative importance of the weights[43]. Various other nonuniform quantization methods have also been adopted by the broader neural networks community to improve efficiency of neural networks[44,45]. Based on all the pulse measurement results, the trilayer RRAM devices show gradual conductance switching in MΩ regime that can overcome the drawbacks of filamentary RRAM devices which show binary resistance states in kΩ regime. We investigated cycling properties of our trilayer bulk RRAM devices by performing endurance measurements based on pulse programming. As shown in Supplementary Fig. S4a, the pulse endurance test results exhibit stable weight modulation until $2 \times 10^5$ cycles under set/reset pulses (Set: $-2.0$ V 5 ms/Reset: 1.0 V 5 ms). We also extracted the variations (σ) from the endurance cycling tests and the variations were about 1% which is enough to differentiate the different conductance states. Read disturbance is tested up to 200k cycles and they show no degradation in device characteristics due to uniform and stable bulk RRAM switching (Supplementary Fig. S4).

## Hardware SNN implementation with RRAM crossbars

For the hardware implementation of neural networks, we first investigated the effect of ON and OFF state resistances ($R_{ON}$ and $R_{OFF}$) on the read and write operations across crossbar arrays using circuit simulations (HSPICE). Supplementary Fig. S5a shows that for $R_{OFF} < ~10$ MΩ, the read margin significantly degrades as the array size increases. For the write operation, the voltage across individual RRAM cells decreases with $R_{ON}$ (Supplementary Fig. S5b). These results indicate the importance of MΩ range resistance to maintain read and write accuracy for selector-less crossbars. Although MΩ resistance and non-linearity of trilayer RRAM are great for reliable crossbar operation, a small dynamic range ($R_{ON}/R_{OFF} ~ 2.5$) is a limiting factor. To address that, we employed a row-differential encoding scheme (Fig. 5a), where two RRAMs represent positive and negative weights by utilizing opposite voltage polarity, i.e., $V_{WL+} = V_{ref} + V_{READ}$, $V_{WL-} = V_{ref} - V_{READ}$. The differential conductance 'Diff_G' given by $G^+$-$G^-$, represents both positive and negative weights. For the differential read of multi-level RRAM, the effective dynamic range depends on the minimum achievable conductance difference ($Diff\_G_{min}$) as in equation '2($G_{max}$-$G_{min}$)/$Diff\_G_{min}$' (Fig. 5a). It results in a significantly higher dynamic range (~170) compared to the non-differential single RRAM scheme (Fig. 5b) due to the small 'Diff_G_{min}', helping with mapping a wider range of real-valued weights. For hardware implementation with RRAM crossbar arrays, we developed a neuromorphic compute-in-memory platform (Fig. 5d). It utilizes a switched capacitor voltage-sensing circuit to avoid the need for current-sensing schemes relying on high-power large-area transimpedance amplifiers (Fig. 5e, f)[46]. We performed read (Fig. 5g) and MVM computations on the trilayer RRAM crossbar and demonstrated the differential scheme can achieve highly linear MVM computation.

For neuromorphic computing at the edge with trilayer bulk RRAM crossbars, we implemented an SNN trained using Evolutionary Optimization for Neuromorphic Systems (EONS) algorithm[47]. In EONS algorithm, randomly generated populations are used as initial seeds for neural network optimizations. The fitness score, a criterion to measure neural network accuracy is assessed in each neural network during the evaluation step. Then, the selected networks through the tournament methods are used to perform reproduction steps where various operators occur (e.g., duplication, crossover, and mutation). In this research, the SNN was specifically trained for a small-scale autonomous racing task[48] using LIDAR sensor data as the input and producing speed and steering angle as the outputs. For the evolutionary training, the fitness function was defined to evaluate the spiking neural network and to encourage behaviors for completing the task without colliding with a wall[48]. The SNN was trained on 5 Formula-1 tracks and tested on an additional 15 tracks (representative tracks are shown in Fig. 6a) (https://github.com/f1tenth/f1tenth_racetracks), performing pruning after the training. The pruned SNN consists of 14 input neurons and 30 output neurons including recurrent connectivity across and within the layers (Fig. 6b). For the hardware demo, SNN weights were quantized into 4-bit precision and mapped onto RRAM arrays according to the row-differential scheme. Experimentally mapped weights onto the crossbar show high consistency with the ideal (target) weight map (Fig. 6c). Figure 6e, f shows steering angle and speed calculated based on experimental RRAM weights in comparison to software simulation during autonomous navigation testing of the Catalunya map. Quantitative comparison of speed and steering angle computations during navigation through all 15 racetracks show great agreement with the ideal software simulation of the SNN (Fig. 6d).

Furthermore, to compare our work to other technologies (i.e., $HfO_x$:Si[49], $HfO_x/TaO_y$[15], and $Al_2O_3/TiO_2$[9]) at the architecture level, we simulated the energy consumption by the SNN model trained for the same navigation tasks while implementing the required synaptic weights using these different RRAM technologies. Supplementary Figs. S6 and S7 show total energy needed to navigate all racetracks and individual

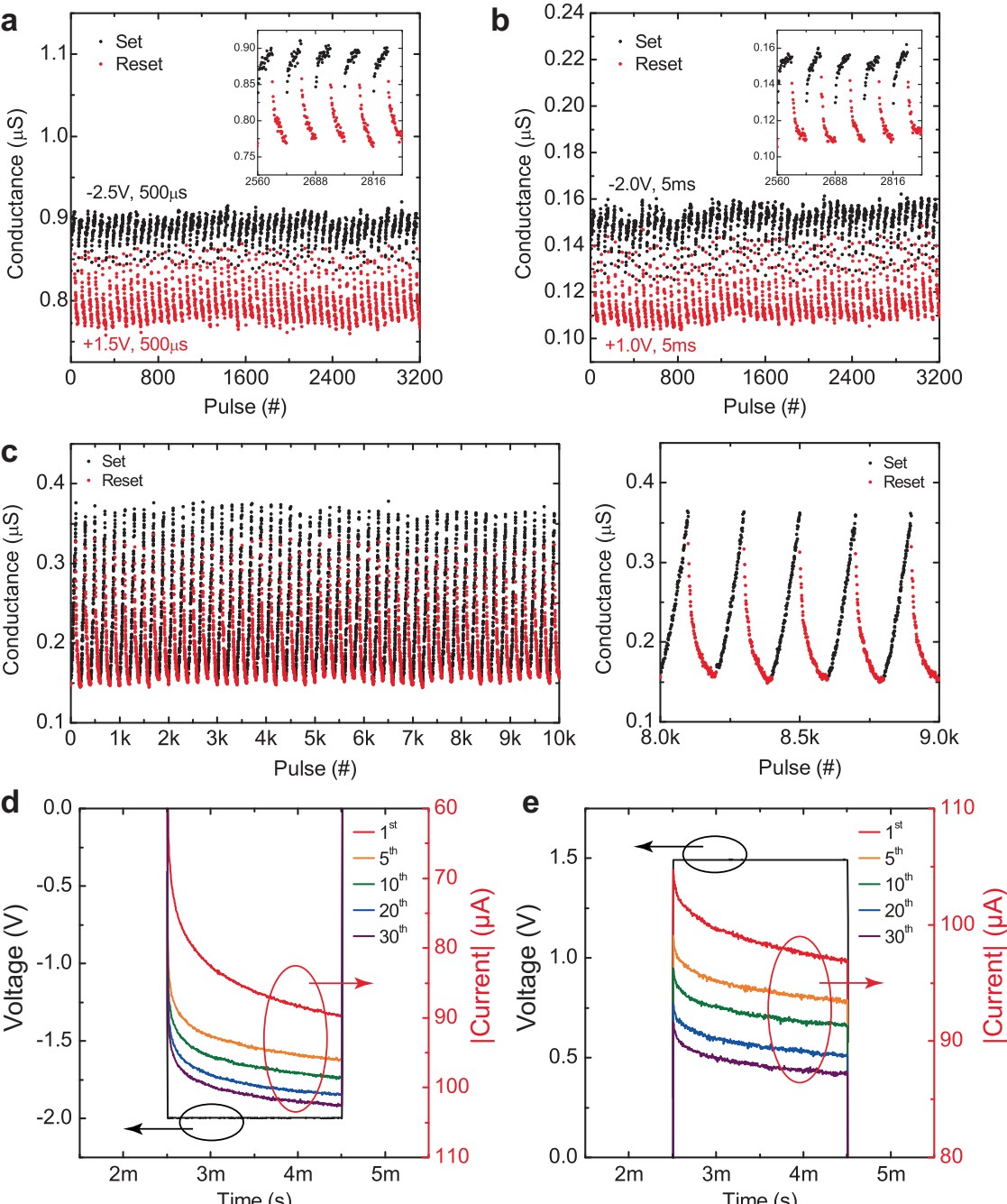

**Fig. 4 | Multilevel gradual switching characteristics of the bulk RRAM devices using pulse measurements. a** Multilevel switching using an identical pulse scheme for 32 different states. Set: −2.5 V, 500 μs/Reset: +1.5 V, 500 μs. **b** Multilevel switching using an identical pulse scheme for 32 different states. Set: −2.0 V, 5 ms/Reset: +1.0 V, 5 ms. **c** Multilevel switching using an incremental pulse scheme for 100 states. Set: −0.8 V to −2.78 V (−20 mV step)/Reset: +0.3 V to +0.993 V (+7 mV step). The transient current measurements using identical pulses **d** set (−2V) and **e** reset (+1.5 V) operations showing multi-level bulk switching without any abrupt current jumps (no filaments).

racetracks, respectively. Our results suggest that our trilayer bulk RRAM substantially (more than order of two) reduces energy consumed by the synaptic arrays in comparisons to other RRAM technologies.

## Discussion

In this work, we successfully demonstrated a forming-free bulk switching RRAM technology by engineering a trilayer metal-oxide stack. We systematically optimized the trilayer oxide stacks which consist of high bandgap tunneling barrier ($Al_2O_3$) and different stoichiometric $TiO_2$ and $TiO_x$ layers. Due to the highly porous and amorphous $TiO_x$ layer, $V_O$ filament formation was effectively suppressed,

whereas the crystalline $TiO_2$ layer showed filamentary switching characteristics. The thickness of the $Al_2O_3$ tunnel barrier was chosen as ~3 nm to set the device resistance to ~MΩ regime (Supplementary Note 3, Supplementary Fig. S8a). Thick $TiO_x$ layer was needed to reduce the electric field across the $V_O$-rich SCLC layer so that facile filament formation due to drift and clustering of $V_O$ could be prevented (Supplementary Fig. S8b). We achieved multi-level, uniform bulk switching in MΩ regime without a compliance current in the bulk RRAM devices.

We benchmarked our bulk RRAM device against different RRAM technologies and showed the advantages on several metrics, i.e.,

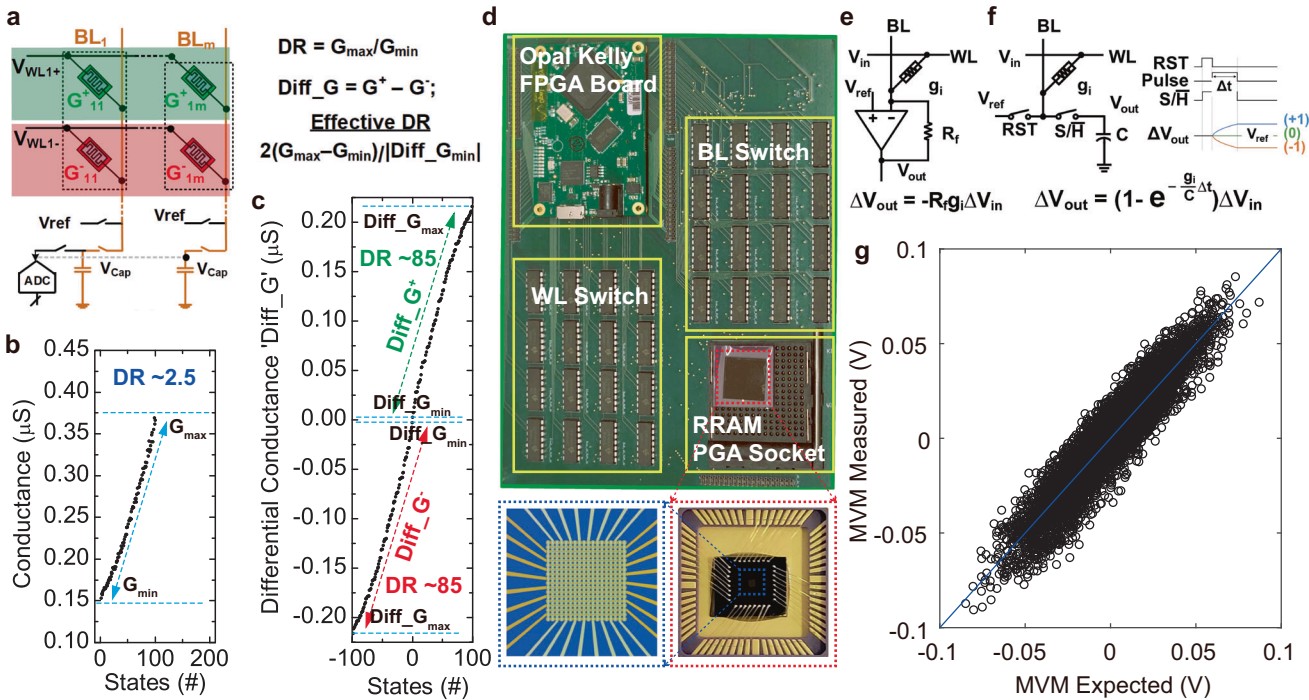

**Fig. 5 | Row-differential voltage sensing using neuromorphic compute-in-memory platform with packaged bulk RRAM crossbar. a** Row-differential scheme. Two RRAMs sharing the same column represent positive and negative weights by applying opposite polarity voltage to respective WLs. **b** Dynamic range for switching for bulk RRAM. **c** Dynamic range enhancement using the row-differential scheme. For 100 levels, the row-differential scheme can increase the effective dynamic range up to ~170. **d** Photograph of neuromorphic compute-in-memory board with packaged bulk RRAM crossbar. **e** Conventional current sensing using transimpedance amplifier. **f** Switched-capacitor voltage-sensing circuit to achieve higher energy efficiency. **g** Measured and expected MVM outputs for the differential encoding.

forming free, multilevel, switching voltages, energy consumption, BEOL compatibility, and robustness (Supplementary Table S2, Supplementary note 4). The key figures of merit of our bulk RRAM technology RRAM technology can be summarized as follows; forming-free operation, CMOS BEOL compatibility, high $R_{on}$ and $R_{off}$ that enables reliable read and write in large scale crossbar arrays and low energy operation, low switching voltages, high number of conductance states, endurance comparable to other RRAM technologies, and much lower total read energy.

Next step for the bulk RRAM technology is scaling device dimensions to the nm regime. One potential concern could be variability for nanoscale devices. For memory technologies, the root cause of the higher variability of characteristics due to cell scaling is related to the number of charged carriers or particles. For instance, the charge trap memory with 10-nm technology node can store only 10 electrons per device which cause severe variability issues in the device characteristics[50]. Therefore, we established a conduction model for our bulk RRAM devices to extract the number of oxygen vacancy defects, which demonstrates $1.26 \times 10^{26}$ $V_O$ defects/m$^3$ in the layer. When we assume the device scales down to 20 nm size with 40 nm thickness, ~2000 $V_O$ defects exist in a single device. The order of $V_O$ defects in a single cell is more than 3 orders higher than the number of trapped electrons in a 10 nm tech node flash memory devices, so variability related to the low number of defects is not expected to be a major problem for our bulk RRAM devices. In addition, the high variability in the conventional RRAM devices come from the stochastic nature of the filaments. Since both set and reset processes are determined by the stochastic movement of atoms in the switching oxides, the device-to-device and cycle-to-cycle variability have been a main problem of filamentary RRAM devices. In our bulk RRAM devices, resistance is modulated by controlling the defect concentration in the switching layer so that the variability problem of stochastic 1D

filaments can be resolved. In addition, the previous study about bulk RRAM devices demonstrated uniform resistive switching characteristics when the device is scaled down to 60×60 nm$^2$ [23]. Finally, for the scaled bulk RRAM devices, the tunnel oxide and the TiO$_x$ switching layer thicknesses could be reoptimized to match MΩ resistance level.

In this work, we developed a neuromorphic CIM platform using bulk RRAM crossbars by combining energy-efficient switched-capacitor voltage sensing circuits with differential encoding of weights. The row-differential weight encoding enabled to increase dynamic range of bulk RRAM devices as well as to give high-accuracy MVM operations. We successfully mapped weights of SNN network for autonomous navigation/racing tasks on Formula-1 racetracks onto bulk RRAM crossbars using the row-differential weight encoding scheme. The fitness score of weight maps on crossbars hardware showed good agreement with ideal software simulation results, suggesting a computational capability of bulk RRAM crossbars. Our work addresses the problems of the filamentary RRAMs and offers a promising pathway towards energy-efficient dynamic on-chip learning with RRAM crossbars.

## Methods
### Bulk RRAM fabrication and packaging crossbars
Ti (12 nm)/Au (100 nm) bottom electrode was deposited by the sputtering on a 4-inch SiO2 (300 nm)/Si wafer with bilayer lift-off process (LOR5B and AZ1512). Plasma-enhanced chemical vapor deposition (PECVD) SiO$_2$ (150 nm) layer was deposited as an insulating interlayer dielectric layer. Various via-hole sizes (Diameter: 3μm to 10μm) were patterned with maskless photolithography and inductively coupled plasma etching process with CF$_4$ atmosphere. The Al$_2$O$_3$/TiO$_2$ (3 nm/3 nm) atomic layer deposition (ALD) layer was deposited with trimethyl aluminum (TMA) and titanium chloride (TiCl$_4$) precursor and water oxidant without breaking vacuum. The sputtered TiO$_x$ layer was

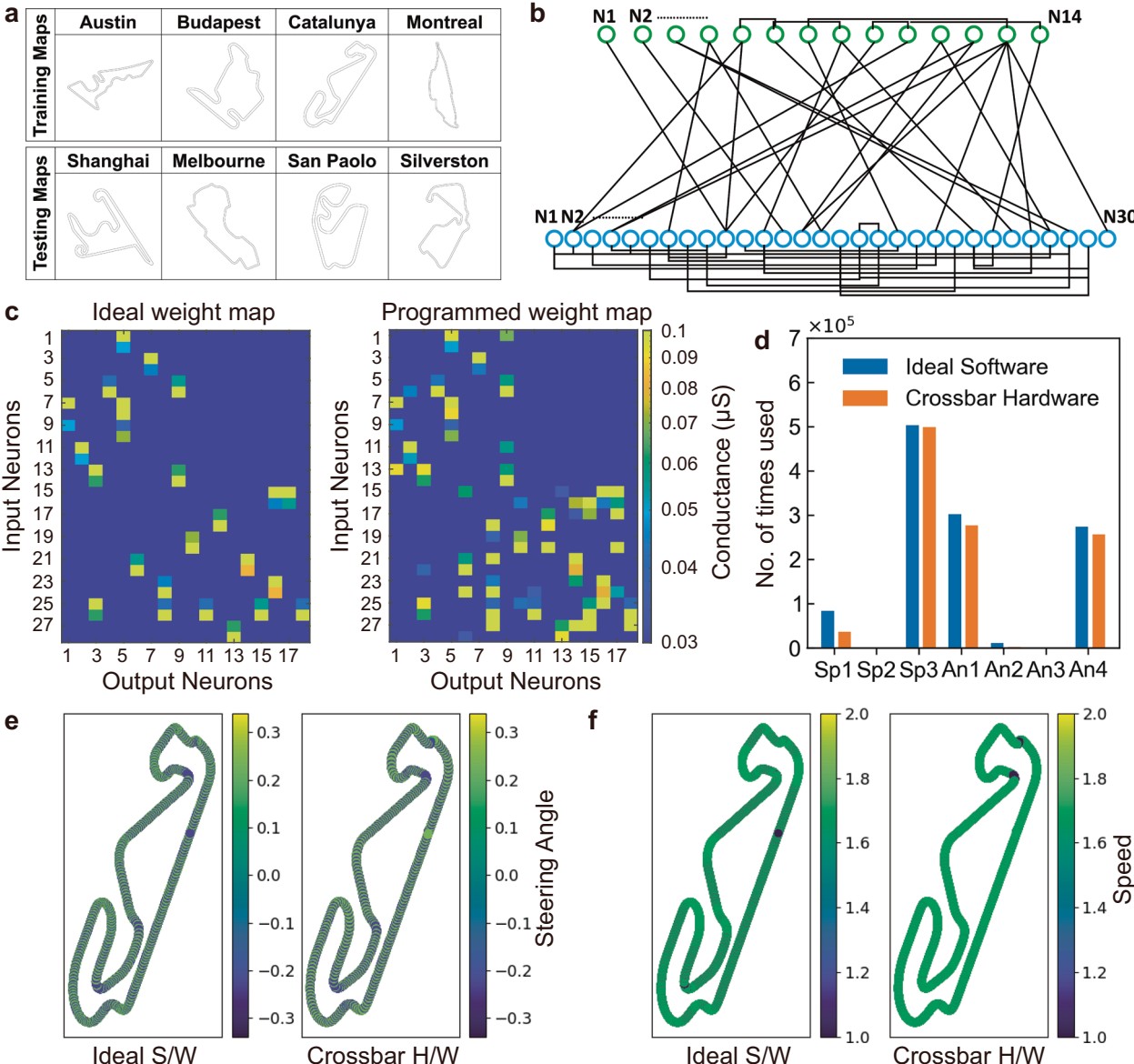

**Fig. 6 | Hardware implementation of SNN for a navigation/racing task.**
**a** Examples of training and testing racetracks for navigation tasks. **b** Schematics of trained SNN (14 input/30 output neurons). **c** Weight map comparison between ideal weights and experimentally programmed weights on crossbars using the row-differential encoding. Two 16×16 crossbars were used for weight mapping. **d** Number of speed (Sp = 1, 1.6, 1.7) and steering angle (An = −0.23, 0, 0.17, 0.23)

computations across navigation through all fifteen racetracks. Ideal software simulation and crossbar hardware implementation show highly consistent results. (Fitness score: 0.54 (Ideal S/W) vs. 0.43 (Crossbar H/W)). Ideal software vs. crossbar hardware computation of **e** steering angle and **f** speed during testing in the Catalunya map.

deposited with sputtering under the different oxygen partial pressures to induce the $V_O$ into the film (S3: 100 W, O₂/(O₂+Ar) = 10%/S4: 200 W, O₂/(O₂+Ar) = 5%). Ti (12 nm)/TiN (22 nm)/Ti (12 nm)/Au (200 nm) top electrode was deposited and patterned analogous to the bottom electrode lift-off process. The switching layer was etched away by plasma etching processed with O₂/CF₄/Ar/BCl₃ gas chemistry. The Au wire bonding was done using manual west bond ball bonder equipment.

**Materials characterization**
For structural characterization, high-resolution X-ray scattering measurements (Grazing Incidence X-ray diffraction and X-ray reflection) were conducted using in-house X-ray diffraction (Smartlab XRD, Rigaku). Transmission electron microscopy (TEM)-ready samples were prepared using the in-situ FIB lift-out technique on an FEI Dual Beam FIB/SEM. The samples were capped with sputtered Ir and e-Pt/I-Pt prior to milling. The TEM lamella thickness was ~100 nm. The samples were

imaged with a FEI Tecnai TF-20 FEG/TEM operated at 200 kV in bright-field (BF) TEM mode, high-resolution (HR) TEM mode, and high-angle annular dark-field (HAADF) STEM mode. The STEM probe size was 1-2 nm nominal diameter.

**Electrical characterization and weight mapping on crossbars**
The electrical I-V characteristics of the RRAM devices were measured using a semiconductor analyzer (4155 C, Agilent) and switching matrix (E5250A, Keysight). A pulse generator unit (81110 A, Agilent) and pulse measurement units with remote amplifiers (4200-SCS with 4225-PMU and 4225-RPM, Keithley) were used for the pulse generation and measurements.

The bulk RRAM crossbar arrays were wire-bonded on pin grid arrays and mounted on custom designed printed circuit board (PCB) to map the weights on the arrays. Weight mapping process on the arrays was conducted using connected switching matrix,

semiconductor analyzer, and a pulse generator unit. We implemented an Opal Kelly FPGA Board to demonstrate the voltage sensing scheme[46]. The conductance was calculated by driving WLs to $V_{pulse}$ and measuring the time constant of BL charging. Then the absolute conductance was calculated by the following expression.

$$V_{BL} \approx V_{ref} + \frac{g_i}{\sum_{k=1}^{N} g_k} V_{pulse}, \text{for } t \gg \frac{C}{\sum_{k=1}^{N} g_k} \quad (1)$$

For row differential MVM read-out to obtain expected versus measured results, we took difference between RRAM devices (G⁺ and G⁻) from two consecutive word-lines (WL⁺ and WL⁻) on the same bit-line (BL). To perform the read-out operation, the ternary inputs (X = [−1, 0, 1]) are assigned to each differential pair. When the input Xi = 1, +Vread (Vref +0.1 V) is applied to the WL⁺ and −Vread (Vref −0.1 V) is applied to the WL⁻. In the case of Xi = −1, −Vread (Vref −0.1 V) is applied to the WL⁺ and +Vread (Vref +0.1 V) is applied to WL⁻. If the input Xi = 0, then Vref is applied to both WL⁺ and WL⁻.

The charged voltage on the sampling capacitor is expressed as Eq. (2).

$$V_{row-diff} = V_{ref} + \frac{\sum_i (G_i^+ - G_i^-) \times X_i \times \left| V_{read} - V_{ref} \right|}{\sum_i (G_i^+ + G_i^-)} \quad (2)$$

where i = [1, 2, …,8] represents the number of differential pairs and Vref is the pre-charge voltage of sampling capacitor.

The expected MVM values are calculated after inserting the extracted conductance values and the random sequence ternary input vector 'Xi' in Eq. (2). In this case, first we read the total conductance value on the shared/selected BL and then individual conductance value of each RRAM on the same BL. To obtain the total conductance, we activated all the WLs with Vread (Vref +0.1 V) and extracted RC time constant of the charged voltage on a known value sampling capacitor at selected BL. After deriving total conductance of the BL, the conductance of each cross point is extracted by applying Vread (Vref +0.1 V) on the targeted WL and Vref on the rest of the WLs. When the charged voltage on the sampling capacitor reaches saturation, the measured voltage is proportional to the ratio of individual RRAM conductance to the total conductance:

$$V = V_{ref} + \frac{G_{target}}{G_{total}} \left| V_{read} - V_{ref} \right| \quad (3)$$

where $G_{total} = \sum_{i=1}^{8} (G_i^+ + G_i^-)$.

On the other hand, to obtain the measured MVM values, we applied a random sequence of ternary input vector 'Xi' to the differential pairs. For every ternary random input vector, we captured the charged voltage on the sampling capacitor using commercial 16-bit resolution successive approximation register analog-to-digital converter (SAR-ADC) (ADS7067, Texas Instruments). We presented these measurement results as expected MVM vs measured MVM which follow the linear trend as shown in paper (Fig. 5g).

**Formula-1 track simulation**

We leveraged the TENNLab Neuromorphic Framework software framework[51], along with Evolutionary Optimization for Neuromorphic Systems (EONS)[47] to design a spiking neural network for evaluation in our hardware. The task that we optimized the neural network for was autonomous control of a small-scale autonomous race car. We leveraged the F1Tenth[52] simulation environment for training. In this environment, the observations provided to the neural network as input are LIDAR observations from the car, and the actions that can be applied (that are produced as output by the network) are steering angle and speed.

We defined discrete values that the network can choose for steering angles ([0, −0.01, 0.01, −0.03, 0.03, −0.05, 0.05, −0.07, 0.07, −0.1, 0.1, −0.13, 0.13, −0.15, 0.15, −0.17, 0.17, −0.2, 0.2, −0.23, 0.23, −0.25, 0.25, −0.27, 0.27, −0.3, 0.3, −0.34, 0.34]) and speed ([1, 1.1, 1.2, 1.3, 1.4, 1.5, 1.6, 1.7, 1.8, 1.9, 2]). Two sets of output neurons are created, one for steering angle and one for speed, and within those sets, one for each legal value. At each step of the simulated environment, the car receives as input 10 LIDAR beams, down-selected from the full 960 LIDAR beams by selecting the maximum beam distance in each of ten equal-sized regions of the 960 beams. Then, the network simulates for 50 time steps, and the output neuron that fires most for steering angle corresponds to the selected steering angle value (and similarly for speed). Observations are taken and actions are applied of 2 milliseconds in the simulation. Thus, the network makes decision about steering angle and speed every 2 milliseconds.

Following methods similar to those in ref. 48, we used EONS to optimize the parameters (synaptic weights and neuron thresholds) and structure (number of hidden neurons and connectivity between neurons) of a single spiking neural network. EONS is an evolutionary algorithm-based approach that begins with an initial population of randomly initialized networks. Then, each network is evaluated to determine a training score. These scores are used with tournament selection to preferentially select better-performing networks to serve as parents. Then, new networks are created from those parents through recombination/crossover and random mutations. The new population is then evaluated, and this process is repeated for a fixed number of generations. In this case, we optimized a single network for 200 generations. The training performance of the network that is used to drive the optimization is the average score across five real-world Formula 1 tracks, where the score for each track is the percentage of two laps completed without crashing. The testing score of the network is the average score across fifteen other Formula 1 tracks (i.e., tracks not used during training). In this previous work, we have seen that the networks trained in simulation are frequently able to translate to successfully operate a small-scale physical autonomous car.

Figure 6d−f shows the direct performance comparison between software and hardware crossbar-based implementation using the speed (Sp = 1, 1.6, 1.7) and steering angle (An = −0.23, 0, 0.17, 0.23) values chosen by the network out of all provided 11 speed values and 29 steering angles. EONS algorithm optimized the structure of the spiking neural network and through the optimization, it determined the most suitable speed and angle values among all the provided speed and angle values needed to perform the task.

## Data availability

The data that support the plots and other results of this paper are available from the corresponding author upon request.

## Code availability

The software codes used for this study are available from the corresponding author upon request.

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

## Acknowledgements
This work was supported by the Office of Naval Research (N000142012405) and the Quantum Materials for Energy Efficient Neuromorphic Computing (Q-MEEN-C), an Energy Frontier Research Center (EFRC), funded by the US Department of Energy, Office of Science, Basic Energy Sciences under award #DE-SC0019273.

## Author contributions
This work was conceived by J.P., A.K., and D.K. Fabrication was performed by J.P., Y.Z., and S.O. Electrical characterization and analysis of RRAM devices were performed by J.P., A.K., and Y.Z. Temperature-dependent electrical characterization were performed by J.P., Y.Z., E.Q. and I.K.S. Materials characterization was performed by J.P. Crossbar analysis was performed by J.P., A.K., Y.S., J.-H.K., S.J., G.H., G.C. SNN Hardware implementation were performed by A.L.N and C.D.S. All the authors discussed the results and contributed to the writing of the manuscript. D.K. supervised the work.

## Competing interests
The authors declare no competing interests.
