## [Peer Review File · Nature Communications]

REVIEWER COMMENTS

Reviewer #1 (Remarks to the Author):

The manuscript, entitled to “Multi-level, forming free, bulk switching trilayer RRAM for neuromorphic computing at the edge”, reported bulk switching operation at megaohm regime with high current nonlinearity using trilayer (Al₂O₃/TiO₂/TiO_x) metal-oxide stacking engineering and demonstrated a neuromorphic compute-in-memory platform based on trilayer bulk RRAM crossbar array. Overall, the authors show the interesting device structures with gradient of oxygen vacancies, which could address challenges posed by existing RRAM technology. Thus, I think this original manuscript deserves to be published in Nature Communication in the final form. But, I have some technical comments on this manuscript, which should be addressed before the publication in Nature Communication.

- 1) In BF-TEM of Fig. 2, the authors showed bulk RRAM stacks with Au/Al₂O₃/TiO₂/TiO_x/Ti/TiN layers. So, it seems that Ti metal layers on top of sputtered (porous) TiO_x layers was partially oxidized based on TEM-EELS spectra, which also influence the concentration gradient of oxygen vacancies. The authors need to address the scientific reason on the oxidation of Ti metal layer.
- 2) The authors claimed that oxygen vacancy formation was effectively suppressed due to the highly porous and amorphous TiO_x layer. I think this is related to the suppression of oxygen vacancy clustering in the highly porous and amorphous TiO_x layer. Please more comment on the scientific origin of bulk switching in terms of oxygen vacancy concentration.
- 3) In Fig.1e-h, the authors showed DC switching curves that do not clearly indicate the switching polarity. Adding arrows to the figures would enhance readability.
- 4) In Fig.4, the authors demonstrated gradual weight updates using both identical and incremental pulses. It appears that the non-linearity of gradual switching was improved using incremental pulses. They should provide more detailed explanations regarding this non-linearity.
- 5) In Fig.5, the authors demonstrated differential voltage sensing scheme using bulk RRAM crossbars. I think that the authors should provide a more detailed description of the method used for reading the devices and the differences between the expected MVM values and the measured MVM values.
- 6) In Fig.6, the authors trained and tested various F1 maps using an optimized SNN by EONS algorithms. However, Fig. 6d only showed 3 speed values and 4 steering angles out of 11 speed values and 29 steering angles. I think that the authors present all the dataset of steering angles and speeds.

Reviewer #2 (Remarks to the Author):

The results reported in the paper have certain merits and novelty, mainly related to (i) the technological development of a forming-free and bulk switching RRAM technology based on a trilayer metal-oxide stack, (ii) a concrete validation using the proposed RRAM by implementing a spiking neural network model for an autonomous navigation/racing task. The field of such RRAM for neuromorphic remains open and competitive and this contribution deserves attention for publication, after clarifying the following aspects:

(1) The authors are making a case that their type of device and technology goes beyond conventional RRAM devices using filamentary switching, which suffers from extensive variations and noise leading to computational accuracy loss and increased energy consumption. However, it is not very clear how much progress is achieved when multiple metrics are compared to state of the art; I would suggest to report in a consolidated table all the key figures of merit of RRAM technology for neuromorphic, including switching voltages, R_{on} , R_{off} , size, programming, robustness, energy consumption, multi-level states, etc. And discuss how it compares and advances the field.

(2) the authors report overlapping DC sweeps over 50 cycles and conclude that the trilayer bulk RRAM exhibits minimal cycle-to-cycle variation. Have the authors looked at much higher order of magnitude of cycling (10^4 - 10^5) and also on any tiny distribution of switching parameters, such as stochastic noise, associated with the operation. Please discuss.

(3) It is not very clear why the "magic" combination of tri-layers with the thicknesses proposed for sample S4 - from the four different multilayer stacks fabricated, S1-S4) - the trilayer with a sputtered TiOx showing bulk switching characteristics.) offer such advantages and control. This looks like a "lucky" combination that was not necessarily predicted by prior calculations or simulations. Can the authors discuss any more way to engineer quantitatively such stacks as material choice and thickness, to achieve similar results and what the optimization method could be? It would be useful to develop a full model to match the device characteristics based on the proposed combination of the conduction mechanism of trilayer bulk RRAM by fitting DC I-V characteristics with direct tunneling, Fowler-Nordheim (FN) tunneling and space-charge-limited conduction (SCLC) models.

(4) Further validation of the electrical characterization proposed, based on the two models should be carried out at various temperatures to fully validate the different temperatures (low and high) dependence of the two supposed mechanisms. If this is confirmed as trends and physical dependences, the conduction mechanism proposed would be greatly confirmed. It may also happen that the multilevel operation is affected by temperature increase and this would be very interesting experiment.

(5) the tested devices have still large size; DC I-V switching curves of trilayer bulk RRAM is tested in cells from $3\mu\text{m}$ to $10\mu\text{m}$. What about maintaining the observed performance and properties when scaling the size of the cells? Any foreseen barriers involving for instance higher variability of characteristics due to cell scaling?

(6) While I find the validation of the hardware implementation of SNN for a navigation/racing task with 16×16 cross bars for weight mapping very interesting and a great results, I am not sure how at architecture level this will compare with other technologies implementing the same. A short discussion with some clear benchmarks of gains would clarify the major advantages beyond the functional aspect.

Reviewer #3 (Remarks to the Author):

The authors report non-filamentary synaptic devices using a trilayer oxide structure to overcome limitations of filamentary memristor devices reported in literature. Using multiple layers including one that is insulating but thin, they argue that filamentary switching can be avoided and show some data for area scaling to support this hypothesis. Comments:

- Please explain the vacancy concentration in the TiO₂ layer and what values are expected in their samples and their distribution to support the claims concerning the conduction mechanism
- Non-filamentary synaptic updates in oxides have indeed been previously reported (Nature Comms, 11, 2245, 2020), please compare and contrast to present results to place work in context of literature.
- It is not clear the difference between S3 and S4 samples why there is a different behavior. What is it about the TiO_x layer that causes this?
- Why is the switching polarity different in S4 compared to other samples, if the same mechanism persists but the concentration of defects is reduced according to the authors?
- Have the authors tried different layer thickness of Al₂O₃ to verify the mechanism?
- If ALD of alumina provides good quality film (ie serves as insulating barrier), why ALD of TiO₂ layer results in low quality samples leading to filamentary switching? Is there some process issue or is this intentional?
- The authors use porous films but clearly they are not controlling the pore location or such. In this scenario, how can the samples be considered reproducible and overcome the variability limitations mentioned in the introduction? Please clarify.

Reviewer #1 (Remarks to the Author):

The manuscript, entitled to “Multi-level, forming free, bulk switching trilayer RRAM for neuromorphic computing at the edge”, reported bulk switching operation at megaohm regime with high current nonlinearity using trilayer (Al₂O₃/TiO₂/TiO_x) metal-oxide stacking engineering and demonstrated a neuromorphic compute-in-memory platform based on trilayer bulk RRAM crossbar array. Overall, the authors show the interesting device structures with gradient of oxygen vacancies, which could address challenges posed by existing RRAM technology. Thus, I think this original manuscript deserves to be published in Nature Communication in the final form. But, I have some technical comments on this manuscript, which should be addressed before the publication in Nature Communication.

Comment 1.

In BF-TEM of Fig. 2, the authors showed bulk RRAM stacks with Au/Al₂O₃/TiO₂/TiO_x/Ti/TiN layers. So, it seems that Ti metal layers on top of sputtered (porous) TiO_x layers was partially oxidized based on TEM-EELS spectra, which also influence the concentration gradient of oxygen vacancies. The authors need to address the scientific reason on the oxidation of Ti metal layer.

Response:

We thank the reviewer for pointing this out. We figured that the composition in the Ti metal layer is TiO_{1.2} based on the composition analysis using STEM-EELS. The reason for partial oxygen contents in the Ti metal layer is the “oxygen scavenging effect” of the Ti. When the Ti metal is deposited onto the TiO₂ layer, the Ti metal is partially oxidized by scavenging oxygen atoms from the underlying TiO₂ layer due to the lower chemical potential of oxygen in Ti suboxides than that in TiO₂.¹ There are previous studies that exploit Ti as a scavenging layer to reduce underlying oxide layers.^{2,3} For example, for the Nb-based selector device fabrication, Ti metal plays an important role in stabilizing the underlying NbO₂ selector layer without further oxidation to the thermally stable Nb₂O₅ composition. The Ti metal also reduces Hf-based oxides to induce oxygen vacancy defects in it so that the RRAM device can form the filaments at the lower set voltage. Therefore, the partial oxidation of the top Ti metal comes from the “scavenging effect” of the Ti metal layer. The conductivity of TiO_{1.2} is expected to be 10³ Ω·m which is many orders higher than the stoichiometric TiO₂ layer⁴, so its effect on the overall device resistance and potential profile across the switching layer would be negligible.

Following the reviewer's comment, we added the following explanations to "**Optimization of trilayer bulk RRAM stack**" section in the manuscript:

"We estimated the composition of Ti metal layer as $\text{TiO}_{1.2}$ based on the composition analysis. The top Ti metal layer scavenges the oxygen from the sputtered TiO_x layer due to the lower chemical potential of oxygen in Ti suboxides than that in TiO_2 .¹ There are previous studies that exploit Ti as a scavenging layer to reduce underlying oxide layers.^{2,3} For example, for the Nb-based selector device fabrication, Ti metal plays an important role in stabilizing the underlying NbO_2 selector layer without further oxidation to the thermally stable Nb_2O_5 composition. The Ti metal also reduces Hf-based oxides to induce oxygen vacancy defects in it so that the RRAM device can form the filaments at the lower set voltage."

Comment 2.

The authors claimed that oxygen vacancy formation was effectively suppressed due to the highly porous and amorphous TiO_x layer. I think this is related to the suppression of oxygen vacancy clustering in the highly porous and amorphous TiO_x layer. Please more comment on the scientific origin of bulk switching in terms of oxygen vacancy concentration.

Response:

We thank the reviewer for pointing this out. In the first sentence, the reviewer probably intended to say that "oxygen vacancy filament formation" is suppressed in the switching layer not "oxygen vacancy formation", as the second sentence mentions oxygen vacancy clustering. We will explain the switching mechanism accordingly.

We would like first to explain the filamentary switching mechanism for a better understanding of switching in bulk RRAM. In the filamentary RRAM, V_O defects are well known to be mobile with external electrical and thermal stimuli.⁵ Filamentary RRAM needs an initial high voltage electroforming step which forms the V_O defect filaments between two electrodes. Once the filaments have formed, bipolar switching takes place due to the forming and rupturing of the filaments. Meanwhile, the grain boundaries are well known to be the high diffusivity paths of small ions such as oxygens or hydrogens in crystalline oxides.^{6,7} Especially for our polycrystalline filamentary RRAM devices (S1, S2), the grain boundaries play an important role in charge transport and V_O accumulation and diffusion. Due to these diffusivity paths, filament formation and rupture occur much easily in the filamentary RRAM devices based on poly crystalline oxides.

In our bulk RRAM devices, we deposited an amorphous, porous, and V_O -rich thick TiO_x layer instead of having a thick crystalline ALD TiO_2 layer. Due to the absence of fast diffusion paths or accumulation sites for V_O defects in the amorphous phase, the V_O defects are not clustered in specific locations that facilitate filament formation. The filament formation is effectively suppressed in an amorphous layer as compared to the crystalline phase in RRAM devices.⁸ V_O defects will drift homogeneously throughout the entire area of the layer rather than forming defects-clustered filaments, following the direction of the electric field, enabling bulk switching instead of filamentary switching. When a positive voltage is applied to the top electrode, the V_O are pushed downwards following the electric field and the V_O concentration in the TiO_x layer is reduced. Space-charge-limited-conduction (SCLC) dominates the conduction in the TiO_x layer. Since the V_O concentration in the TiO_x layer is reduced, the SCLC current decreases confirming that the device is reset to a higher resistance state. We successfully confirmed this model by fitting IV characteristics at different resistance states to our model including tunneling and SCLC conduction (**Fig. 3g**) and extracted the trap density (**Fig. 3h**). The trap density is decreased after the device is programmed to higher resistance states using positive bias.

In response to this comment, we added explanations below to the “**Characterization of bulk RRAM switching behavior**” section in the manuscript:

“Bulk switching can be better understood by reviewing filamentary RRAM first. In the filamentary RRAM, V_O defects are well known to be mobile with external electrical and thermal stimuli.⁵ Filamentary RRAM needs an initial high voltage electroforming step which forms the V_O defect filaments between two electrodes. Once the filaments have formed, bipolar switching takes place due to the forming and rupturing of the filaments. Meanwhile, the grain boundaries are well known to be the high diffusivity paths of small ions such as oxygens or hydrogens in crystalline oxides.^{6,7} Especially for our polycrystalline filamentary RRAM devices (S1, S2), the grain boundaries play an important role in charge transport and V_O accumulation and diffusion. Due to these diffusivity paths, filament formation and rupture easily occur in the filamentary RRAM devices.

In our bulk RRAM devices, we deposited an amorphous, porous, and V_O -rich thick TiO_x layer instead of having a crystalline ALD TiO_2 layer. Due to the absence of fast diffusion paths or accumulation sites for V_O defects in the amorphous phase, the V_O defects are not clustered in specific locations that facilitate filament formation. The filament formation is effectively suppressed in an amorphous layer as compared to the crystalline phase in RRAM

devices.⁸ V_O defects will drift homogeneously throughout the entire area of the layer rather than forming defects-clustered filaments, following the direction of the electric field, enabling bulk switching instead of filamentary switching. When a positive voltage is applied to the top electrode, the V_O are pushed downwards following the electric field and the V_O concentration in the TiO_x layer is reduced. Space-charge-limited-conduction (SCLC) dominates the conduction in the TiO_x layer. Since the V_O concentration in the TiO_x layer is reduced, the SCLC current decreases confirming that the device is reset to a higher resistance state. We successfully confirmed this model by fitting IV characteristics at different resistance states to our model including tunneling and SCLC conduction (**Fig. 3g**) and extracted the trap density (**Fig. 3h**). The trap density is decreased after the device is programmed to higher resistance states using positive bias.”

Comment 3.

In Fig. 1e-h, the authors showed DC switching curves that do not clearly indicate the switching polarity. Adding arrows to the figures would enhance readability.

Response:

We thank the reviewer for catching this. We think the arrow got deleted when we group the figures for the final submission. We now corrected this mistake and added the arrows in figure 2.

Comment 4.

In Fig.4, the authors demonstrated gradual weight updates using both identical and incremental pulses. It appears that the non-linearity of gradual switching was improved using incremental pulses. They should provide more detailed explanations regarding this non-linearity.

Response:

We thank the reviewer for a great suggestion. As the reviewer mentioned, linear and symmetric switching characteristics are desirable particularly for online learning and on-chip training applications. However, achieving perfectly linear and symmetric weight updates are extremely challenging to achieve with RRAM devices. Following, the reviewer's suggestion, we investigated non-linearity of the trilayer bulk RRAM devices using the following equations.⁹

$$G_{LTP} = B \left(1 - e^{\left(\frac{-P}{A}\right)} \right) + G_{min}, G_{LTD} = -B \left(1 - e^{\left(\frac{P-P_{max}}{A}\right)} \right) + G_{max}$$

$$B = (G_{max} - G_{min}) / \left(1 - e^{\frac{-P_{max}}{A}} \right)$$

We demonstrated weight updates using both identical and incremental pulse schemes. When we adopt the identical pulse scheme, the non-linearity values of +3.68 and -4.34 (**Fig. S3a**) during the potentiation and depression were achieved. The non-linearity could be improved to -1.24 and -4.21 (**Fig. S3b**) for the potentiation and depression process by using the incremental pulse scheme. We expect that the non-linearity could be further improved by optimizing the pulse amplitude and width for potentiation and depression in the incremental scheme. Furthermore, there are algorithm-device co-design approaches that can be adopted to compensate system-level effects of switching nonlinearity. For instance, to mitigate the non-linearity effect on accuracy drop in online learning and classification tasks, we previously developed the adaptive quantization method, which maps weights onto the device conductances based on the distribution and relative-importance of the weights.¹⁰ Various other nonuniform quantization methods have also been adopted by the broader neural networks community to improve efficiency of neural networks.^{11,12}

In response to this comment, we added the below explanations to the “**Characterization of bulk RRAM switching behavior**” section in the manuscript and the supplementary information:

“We quantitatively analyzed the device non-linearity and found that incremental pulse scheme can improve non-linearity (**Fig. S3, Supplementary note 2**). The non-linearity could be improved by further optimizing the pulse amplitude and width for potentiation and depression. To compensate for the non-linearity effect in hardware implementation of neural

networks, we previously developed the adaptive quantization method, which maps neural network weights onto the device conductances based on the distribution and relative importance of the weights.¹⁰ Various other nonuniform quantization methods have also been adopted by the broader neural networks community to improve efficiency of neural networks.^{11,12}”

Below explanations are added to **Supplementary note 2** and **figures S3** in the supplementary information:

“To quantitatively analyze the device non-linearity, we investigated non-linearity of the trilayer bulk RRAM devices using the following equations.⁹

$$G_{LTP} = B \left(1 - e^{\left(\frac{-P}{A}\right)} \right) + G_{min}, G_{LTD} = -B \left(1 - e^{\left(\frac{P-P_{max}}{A}\right)} \right) + G_{max} \quad (S8)$$

$$B = (G_{max} - G_{min}) / (1 - e^{\frac{-P_{max}}{A}}) \quad (S9)$$

We demonstrated weight updates using both identical and incremental pulse schemes. When we adopt the identical pulse scheme, the non-linearity values of +3.68 and -4.34 (**Fig. S3a**) during the potentiation and depression were achieved. The non-linearity could be improved to -1.24 and -4.21 (**Fig. S3b**) for the potentiation and depression process by using the incremental pulse scheme. We expect that the non-linearity could be further improved by optimizing the pulse amplitude and width for potentiation and depression in the incremental scheme. Furthermore, there are algorithm-device co-design approaches that can be adopted to compensate system-level effects of switching nonlinearity. For instance, to mitigate the non-linearity effect on accuracy drop in online learning and classification tasks, we previously developed the adaptive quantization method, which maps weights onto the device conductances based on the distribution and relative-importance of the weights.¹⁰ Various other nonuniform quantization methods have also been adopted by the broader neural networks community to improve efficiency of neural networks.¹⁰”

Figure S3. Normalized conductance vs. Normalized pulse using **a.** identical and **b.** incremental pulse schemes.

Comment 5.

In Fig.5, the authors demonstrated differential voltage sensing scheme using bulk RRAM crossbars. I think that the authors should provide a more detailed description of the method used for reading the devices and the differences between the expected MVM values and the measured MVM values.

Response:

We thank the reviewer for this constructive comment.

For row differential MVM read-out using ternary input, we take difference between RRAM devices (G^+ and G^-) from two consecutive word-lines (WL^+ and WL^-) on the same bit-line (BL). To perform the read-out operation, the ternary inputs ($X = [-1, 0, 1]$) are assigned to each differential pair. When the input $X_i = 1$, $+V_{read}$ ($V_{ref}+0.1V$) is applied to the WL^+ and $-V_{read}$ ($V_{ref}-0.1V$) is applied to the WL^- . In the case of $X_i = -1$, $-V_{read}$ ($V_{ref}-0.1V$) is applied to the WL^+ and $+V_{read}$ ($V_{ref}+0.1V$) is applied to WL^- . If the input $X_i = 0$, then V_{ref} is applied to both WL^+ and WL^- . Considering the current contribution from all WLS, the charged voltage on the sampling capacitor can be expressed as equation (2).

$$V_{row-diff} = V_{ref} + \frac{\sum_i(G_i^+ - G_i^-) \times X_i \times |V_{read} - V_{ref}|}{\sum_i(G_i^+ + G_i^-)} \quad (2)$$

where $i = [1, 2, \dots, 8]$ represents the number of differential pairs and V_{ref} is the pre-charge voltage of sampling capacitor.

The **expected MVM** values are calculated after inserting the extracted conductance values and the random sequence ternary input vector ‘ X_i ’ in equation (2). In this case, first we read the total conductance value on the shared/selected BL and then individual conductance value of each RRAM on the same BL. To obtain the total conductance, we activated all the WLS with V_{read} ($V_{ref} + 0.1V$) and extracted RC time constant of the charged voltage on a known value sampling capacitor at selected BL. After deriving total conductance of the BL, the conductance of each cross point is extracted by applying V_{read} ($V_{ref} + 0.1V$) on the targeted WL and V_{ref} on the rest of the WLS. When the charged voltage on the sampling capacitor reaches saturation, the measured voltage is proportional to the ratio of individual RRAM conductance to the total conductance:

$$V = V_{ref} + \frac{G_{target}}{G_{total}} |V_{read} - V_{ref}| \quad (3)$$

where $G_{total} = \sum_{i=1}^8 (G_i^+ + G_i^-)$.

On the other hand, to obtain the **measured MVM values**, we applied a random sequence of ternary input vector ‘Xi’ to the differential pairs. For every ternary random input vector, we captured the charged voltage on the sampling capacitor using commercial 16-bit resolution successive approximation register analog-to-digital converter (SAR-ADC) (ADS7067, Texas Instruments). We presented these measurement results as expected MVM vs measured MVM which follow the linear trend as shown in paper (**Fig. 5g**).

The difference between the expected MVM and the measured MVM values arises from non-ideal factors, including potential sneak paths from neighboring columns when most devices are programmed into low-R state, I-R (Interconnect Resistance) drop along the BLs and WLs. Also, the switches used for driving the WLs and BLs have non-zero resistances which further contributes to the non-linearity on the output. This might induce errors when we extract the individual conductance. Moreover, the inactivated devices in the column or BL contribute to the sampled capacitor voltage during the single device conductance measurement. Such contributions from the inactive devices might be reflected in the extracted single device conductance, G_{target} , in equation (3), which further introduces calculation errors in the expected MVM value. Thus, all these factors contribute to the deviation between expected and measured MVM.

For better clarity, we have added the following lines to the methods “**Electrical Characterization and Weight Mapping on Crossbar**” section of the paper.

“For row differential MVM read-out to obtain expected versus measured results, we took difference between RRAM devices (G^+ and G^-) from two consecutive word-lines (WL^+ and WL^-) on the same bit-line (BL). To perform the read-out operation, the ternary inputs ($X = [-1, 0, 1]$) are assigned to each differential pair. When the input $X_i = 1$, $+V_{read}$ ($V_{ref}+0.1V$) is applied to the WL^+ and $-V_{read}$ ($V_{ref}-0.1V$) is applied to the WL^- . In the case of $X_i = -1$, $-V_{read}$ ($V_{ref}-0.1V$) is applied to the WL^+ and $+V_{read}$ ($V_{ref}+0.1V$) is applied to WL^- . If the input $X_i = 0$, then V_{ref} is applied to both WL^+ and WL^- .

The charged voltage on the sampling capacitor is expressed as equation (2).

$$V_{row-diff} = V_{ref} + \frac{\sum_i(G_i^+ - G_i^-) \times X_i \times |V_{read} - V_{ref}|}{\sum_i(G_i^+ + G_i^-)} \quad (2)$$

where $i = [1, 2, \dots, 8]$ represents the number of differential pairs and V_{ref} is the pre-charge voltage of sampling capacitor.

The expected MVM values are calculated after inserting the extracted conductance values and the random sequence ternary input vector ‘ X_i ’ in equation (2). In this case, first we read the total conductance value on the shared/selected BL and then individual conductance value of each RRAM on the same BL. To obtain the total conductance, we activated all the WLs with V_{read} ($V_{ref} + 0.1V$) and extracted RC time constant of the charged voltage on a known value sampling capacitor at selected BL. After deriving total conductance of the BL, the conductance of each cross point is extracted by applying V_{read} ($V_{ref} + 0.1V$) on the targeted WL and V_{ref} on the rest of the WLs. When the charged voltage on the sampling capacitor reaches saturation, the measured voltage is proportional to the ratio of individual RRAM conductance to the total conductance:

$$V = V_{ref} + \frac{G_{target}}{G_{total}} |V_{read} - V_{ref}| \quad (3)$$

where $G_{total} = \sum_{i=1}^8 (G_i^+ + G_i^-)$.

On the other hand, to obtain the **measured MVM values**, we applied a random sequence of ternary input vector ‘ X_i ’ to the differential pairs. For every ternary random input vector, we captured the charged voltage on the sampling capacitor using commercial 16-bit resolution successive approximation register analog-to-digital converter (SAR-ADC) (ADS7067, Texas Instruments). We presented these measurement results as expected MVM vs measured MVM which follow the linear trend as shown in paper (**Fig. 5g**).”

Furthermore, we added the following discussion to the supplementary file as **Supplementary note5**:

“The difference between the expected MVM and the measured MVM values arises from non-ideal factors, including potential sneak paths from neighboring columns, I-R (Interconnect Resistance) drop along the BLs and WLs. Also, the switches used for driving the WLs and BLs have non-zero resistances which further contributes to the non-linearity on the output. This might induce errors when we extract the individual conductance. Moreover, the

inactivated devices in the column or BL contribute to the sampled capacitor voltage during the single device conductance measurement. Such contributions from the inactive devices might be reflected in the extracted single device conductance, G_{target} , in equation (3) in method, which further introduces calculation errors in the expected MVM value. Thus, all these factors contribute to the deviation between expected and measured MVM.”

Comment 6.

In Fig.6, the authors trained and tested various F1 maps using an optimized SNN by EONS algorithms. However, Fig. 6d only showed 3 speed values and 4 steering angles out of 11 speed values and 29 steering angles. I think that the authors present all the dataset of steering angles and speeds.

Response:

We thank the reviewer for comments and apologies for not clarifying it. Reviewer pointed out correctly that there are more number of speed and angle values available to the network. However, we want to clarify here that the mentioned speed ($S_p = 1, 1.6, 1.7$) and steering angle ($A_n = -0.23, 0, 0.17, 0.23$) values in Figure 6d are the only ones chosen by the network itself to perform the specific navigation task. Other speed and angles weren't required by the network and remained unused while performing this navigation task. This happened because EONS algorithm optimized the structure of the network and during the optimization, it determined the optimal speed/angle values to perform the task. For different types of navigation tasks with increasing complexity or constraints, EONS training could prefer a wider range of angle and speed configurations.

For better clarity, we have added the following lines to the methods “**Formula-1 track simulation**” section of the paper.

“**Fig. 6d, 6e and 6f** show the direct performance comparison between software and hardware crossbar-based implementation using the speed ($S_p = 1, 1.6, 1.7$) and steering angle ($A_n = -0.23, 0, 0.17, 0.23$) values chosen by the network out of all provided 11 speed values and 29 steering angles. EONS algorithm optimized the structure of the spiking neural network and through the optimization, it determined the most suitable speed and angle values among all the provided speed and angle values needed to perform the task.”

Reviewer #2 (Remarks to the Author):

The results reported in the paper have certain merits and novelty, mainly related to (i) the technological development of a forming-free and bulk switching RRAM technology based on a trilayer metal-oxide stack, (ii) a concrete validation using the proposed RRAM by implementing a spiking neural network model for an autonomous navigation/racing task. The field of such RRAM for neuromorphic remains open and competitive and this contribution deserves attention for publication, after clarifying the following aspects:

Comment 1.

The authors are making a case that their type of device and technology goes beyond conventional RRAM devices using filamentary switching, which suffers from extensive variations and noise leading to computational accuracy loss and increased energy consumption. However, it is not very clear how much progress is achieved when multiple metrics are compared to state of the art; I would suggest to report in a consolidated table all the key figures of merit of RRAM technology for neuromorphic, including switching voltages, Ron, Roff, size, programming, robustness, energy consumption, multi-level states, etc. And discuss how it compares and advances the field.

Response:

We thank the reviewer for a great suggestion. We have made a consolidated benchmark table based on previous RRAM research for neuromorphic applications. There are many different RRAM technologies but here we focused on publications that demonstrate array scale implementation for both filamentary and bulk RRAM technologies.

Table S2. Benchmarking of conventional RRAM technologies for neuromorphic applications.

RRAM Technologies	Al ₂ O ₃ /TiO ₂ [13]	HfO _x /TaO _y [14]	HfO ₂ :Si [15]	PCMO [16]	NdNiO ₃ [17]	Al ₂ O ₃ /TiO ₂ /TiO _x [This work]
Switching Type	Filamentary	Filamentary	Filamentary	Bulk	Bulk	Bulk
Forming voltage	~3.3V	~4.0V	~2.0V	None	None	None
CMOS Compatibility	✓	✓	✓	X	X	✓
Ron	16 KΩ	23.8 KΩ	6 KΩ	6.25 MΩ	2 KΩ	2.78 MΩ
Roff	333 KΩ	500 KΩ	600 KΩ	157 MΩ	4.5 KΩ	6.67 MΩ
Cell area (μm ²)	0.0625 μm ²	0.360 μm ²	0.04 μm ²	0.0225 μm ²	70700 μm ²	19.6 μm ²
Switching voltage	1.2 V / -1.4 V	+1.6 V / -1.9 V	+1.7 V / -2.0 V	-3.0V / +3.0V	-3.0 V / 5.4 V	-1.8V / + 0.65V
Pulse width	1ms	10us	100ns	1ms	960ns	5ms
# of states	16	8	2	50	32	100
Endurance	10 ⁶	10 ⁶	10 ⁵	10 ⁴	-	2*10 ⁵
Total Energy per read operations	51.0 fJ	44.02 fJ	167.7 fJ	1.04 fJ	722.2 fJ	~ 0.51 fJ

In the supplementary table S2 shown above, the left three columns compares relatively mature filamentary RRAM technologies that consist of widely used RRAM materials HfO_x , TaO_x , AlO_x , or TiO_x .¹³⁻¹⁵ Filamentary RRAM devices need an initial high voltage forming step ($V > 2.0\text{V}$) to generate a conductive filament that is not compatible with advanced CMOS technology nodes. When the filaments are formed through the switching layer, the device switches to a low resistance state (LRS) and shows a few $\text{K}\Omega$ of LRS resistance (R_{ON}). The low R_{ON} not only constrains the array size and number of parallel MAC operations but also increases the read energy consumption and limits the voltage drop across the selected causing unsuccessful write operations as shown in **Figure S5b**. Furthermore, filamentary RRAM devices need complex read and verify schemes to program the devices to the target conductance states which substantially increases the overall energy consumption and limits the online learning capability using these devices. In addition, filamentary RRAM devices can achieve a small number of states, providing precision less than 4-bit which is not sufficient for most neuromorphic computing applications.

Right three columns compare recently developed bulk RRAM devices based on NdNiO_3 , $\text{Pr}_{0.7}\text{Ca}_{0.3}\text{MnO}_3$ and $\text{Al}_2\text{O}_3/\text{TiO}_2/\text{TiO}_x$ to address the limitations of filamentary RRAM devices explained above.^{16,17} First, these bulk RRAM technologies do not need any initial high voltage forming step. The resistance modulation occurs in $\text{M}\Omega$ regime ensuring stable and reliable large size array operations. The cell size of these bulk RRAM is larger than the filamentary RRAM since the technology is less mature and has not yet translated into foundry scale fabrication in contrast to HfO_x based filamentary RRAM shown in the table. Since all bulk RRAM devices exhibit higher resistance than filamentary RRAM, the energy consumption is much lower for the read operation. In addition, many number of states could be achieved to represent the synaptic weights in neuromorphic computing applications. Among the three bulk RRAM technologies, only our work presents a large array-scale demonstration. Furthermore, perovskite-based bulk RRAM technologies need high process temperature ($T > 500^\circ\text{C}$) so further process development to enable fabrication at low temperatures ($T < 400^\circ\text{C}$) will be necessary for CMOS BEOL compatibility. Our bulk RRAM devices based on $\text{Al}_2\text{O}_3/\text{TiO}_2$ materials not only address the limitations of the mature filament RRAM technologies but are also compatible with integration at the CMOS BEOL to enable a high-density 3D compute-in-memory platform for neuromorphic applications. Based on **Table S2**, the key figures of merit of our bulk RRAM technology RRAM technology can be summarized as follows; forming-free operation, CMOS BEOL compatibility, high R_{on} and R_{off} that enables

reliable read and write in large scale crossbar arrays and low energy operation, low switching voltages, high number of conductance states, endurance comparable to other RRAM technologies, and much lower total read energy.

In response to this comment, we added the below explanations in the **supplementary note 4**:

“Left three columns of the table compares relatively mature filamentary RRAM technologies that consist of widely used RRAM materials HfO_x , TaO_x , AlO_x , or TiO_x .¹³⁻¹⁵ Filamentary RRAM devices need an initial high voltage forming step ($V > 2.0\text{V}$) to generate a conductive filament that is not compatible with advanced CMOS technology nodes. When the filaments are formed through the switching layer, the device switches to a low resistance state (LRS) and shows a few $\text{K}\Omega$ of LRS resistance (R_{ON}). The low R_{ON} not only constrains the array size and number of parallel MAC operations but also increases the read energy consumption and limits the voltage drop across the selected causing unsuccessful write operations as shown in **Fig. S5b**. Furthermore, filamentary RRAM devices need complex read and verify schemes to program the devices to the target conductance states which substantially increases the overall energy consumption and limits the online learning capability using these devices. In addition, filamentary RRAM devices can achieve a small number of states, providing precision less than 4-bit which is not sufficient for most neuromorphic computing applications.

Right three columns compare recently developed bulk RRAM devices based on NdNiO_3 , $\text{Pr}_{0.7}\text{Ca}_{0.3}\text{MnO}_3$ and $\text{Al}_2\text{O}_3/\text{TiO}_2/\text{TiO}_x$ to address the limitations of filamentary RRAM devices explained above.^{16,17} First, these bulk RRAM technologies do not need any initial high voltage forming step. The resistance modulation occurs in $\text{M}\Omega$ regime ensuring stable and reliable large size array operations. The cell size of these bulk RRAM is larger than the filamentary RRAM since the technology is less mature and has not yet translated into foundry scale fabrication in contrast to HfO_x based filamentary RRAM shown in the table. Since all bulk RRAM devices exhibit higher resistance than filamentary RRAM, the energy consumption is much lower for the read operation. In addition, many number of states could be achieved to represent the synaptic weights in neuromorphic computing applications. Among the three bulk RRAM technologies, only our work presents a large array-scale demonstration. Furthermore, perovskite-based bulk RRAM technologies need high process temperature ($T > 500^\circ\text{C}$) so further process development to enable fabrication at low temperatures ($T < 400^\circ\text{C}$) will be necessary for CMOS BEOL compatibility. Our bulk RRAM devices based on $\text{Al}_2\text{O}_3/\text{TiO}_2$ materials not only address the limitations of the mature filament RRAM

technologies but are also compatible with integration at the CMOS BEOL to enable a high-density 3D compute-in-memory platform for neuromorphic applications. Based on **Table S2**, the key figures of merit of our bulk RRAM technology RRAM technology can be summarized as follows; forming-free operation, CMOS BEOL compatibility, high R_{on} and R_{off} that enables reliable read and write in large scale crossbar arrays and low energy operation, low switching voltages, high number of conductance states, endurance comparable to other RRAM technologies, and much lower total read energy.”

We have also added the following text to the “**Discussion**” section in the manuscript to address the reviewer comment.

“We benchmarked our bulk RRAM device against different RRAM technologies and showed the advantages on several metrics, i.e., forming free, multilevel, switching voltages, energy consumption, BEOL compatibility, and robustness (**Table S2, Supplementary note 4**). The key figures of merit of our bulk RRAM technology RRAM technology can be summarized as follows; forming-free operation, CMOS BEOL compatibility, high R_{on} and R_{off} that enables reliable read and write in large scale crossbar arrays and low energy operation, low switching voltages, high number of conductance states, endurance comparable to other RRAM technologies, and much lower total read energy.”

Comment 2.

The authors report overlapping DC sweeps over 50 cycles and conclude that the trilayer bulk RRAM exhibits minimal cycle-to-cycle variation. Have the authors looked at much higher order of magnitude of cycling (10^4 - 10^5) and also on any tiny distribution of switching parameters, such as stochastic noise, associated with the operation. Please discuss.

Response: We thank the review for this comment. **Fig. 1h** shows 50 DC sweep cycles of bulk RRAM devices mainly to point out switching uniformity. The 50 DC sweeps almost perfectly overlapped, while the filamentary RRAM devices show extensive variations from sweep to sweep. We investigated cycling properties of our trilayer bulk RRAM devices by performing endurance measurements based on pulse programming. As shown in **Fig. S4**, the pulse endurance test results exhibit stable weight modulation until 2×10^5 cycles under set/reset pulses (Set: -2.0V 5ms / Reset: 1.0V 5ms). We also extracted the variations (σ) from the endurance cycling tests and the variations were about 1% which is enough to differentiate the different conductance states.

In response to this comment, we added the below explanations to the “**Characterization of bulk RRAM switching behavior**” section in the manuscript:

“We investigated cycling properties of our trilayer bulk RRAM devices by performing endurance measurements based on pulse programming. As shown in **Fig. S4**, the pulse endurance test results exhibit stable weight modulation until 2×10^5 cycles under set/reset pulses (Set: -2.0V 5ms / Reset: 1.0V 5ms). We also extracted the variations (σ) from the endurance cycling tests and the variations were about 1% which is enough to differentiate the different conductance states.”

Figure S4. **a.** Endurance and **b.** read noise during the endurance tests up to 2×10^5 pulses.

Comment 3.

It is not very clear why the "magic" combination of tri-layers with the thicknesses proposed for sample S4 - from the four different multilayer stacks fabricated, S1-S4) - the trilayer with a sputtered TiOx showing bulk switching characteristics.) offer such advantages and control. This looks like a "lucky" combination that was not necessarily predicted by prior calculations or simulations. Can the authors discuss any more way to engineer quantitatively such stacks as material choice and thickness, to achieve similar results and what the optimization method could be? It would be useful to develop a full model to match the device characteristics based on the proposed combination of the conduction mechanism of trilayer bulk RRAM by fitting DC I-V characteristics with direct tunneling, Fowler-Nordheim (FN) tunneling and space-charge-limited conduction (SCLC) models.

Response:

We thank the reviewer for this insightful comment.

We would like to start with how we choose the Al₂O₃ tunnel barrier thickness. As described in the manuscript, current conduction occurs through direct tunneling, FN tunneling, and SCLC models. Based on these models, we simulated the current density under the electric

field with various tunneling barrier (Al_2O_3) thickness layers. We varied tunneling barrier thickness from 20 Å to 40 Å, and the current density of them were plotted in **Fig. S8a**. As the current density in the direct tunneling exponentially decays with the oxide thickness, we expect the current density to be decreased by around 3 orders of magnitude per 1 nm Al_2O_3 thickness. These simulation results suggest that to set the device resistance to in $\sim\text{M}\Omega$ regime, the Al_2O_3 tunneling barrier thickness should be chosen $\sim 30\text{Å}$. So, we have decided 30Å of Al_2O_3 tunneling barrier to make our devices in $\sim\text{M}\Omega$ regime. We plotted the current density of an experimentally measured RRAM device with 30Å tunneling barrier thickness and 5 μm diameter (shown with open circle in **Fig. S8a**) showing 1M Ω resistance at 100 mV read voltage, showing great consistency with predictions based on the tunneling current calculations. Our methodology suggests that there is more room to modulate the barrier thickness depending on the target resistance and the device size.

Once the tunnel barrier thickness is fixed, we systematically optimized the thickness of SCLC conduction layer made of TiO_x . Thinner TiO_x will result in higher electric field across the SCLC layer for the same applied voltage and hence will increase the chances clustering of oxygen vacancies through drift to form filaments. To observe this, we chose two different TiO_x thicknesses (6.5 nm for S3 and 40 nm for S4) leading to a difference in the applied electric field across the SCLC layer. We fitted the J-V data of both S3 and S4 using tunneling and SCLC conduction models, and we extracted the electric field applied across the TiO_x switching layer shown in **Fig. S8b** below. Since S3 has a thinner sputtered layer than S4, it causes a higher electric field across the TiO_x switching layer. The electroforming step in filamentary RRAM devices corresponds to the controlled soft-breakdown process in a thin insulator film.^{18,19} Therefore, the thin oxide film or high electric field are known to induce facile filament formation behavior in the RRAM devices. Due to high electric field in S3, it causes filament forming under a high voltage regime ($|V| > 1\text{V}$) as seen in Fig. 1g in the manuscript because the V_O filaments protrude the whole switching layer. This filamentary switching mechanism is the same as the ALD bilayer filamentary RRAM devices (S1 and S2). In a lower voltage regime ($|V| < 1\text{V}$) where electric field is not large enough to drift and cluster V_O , it shows bulk switching behavior through modulation of V_O distribution across the switching layer. In S4, however, we successfully suppressed the filament formation by reducing the electric field with a thick and amorphous TiO_x layer so that we could achieve stable bulk switching behavior.

Figure S8. a. Current density versus applied voltage across the RRAM switching layer. The Al_2O_3 thickness was varied from 20Å to 40Å to determine the experimental Al_2O_3 thickness. Open circle curve shows experimentally measured current density from an RRAM device with 30Å tunneling barrier thickness and 5 μm diameter, exhibiting 1M Ω resistance at 100mV read voltage. **b.** Electric field applied across the TiO_x sputtered layer in S3 and S4.

Following the reviewer’s comment, we added the following explanations to the **Supplementary Note 3** section in the supplementary information:

“The current conduction occurs through direct tunneling, FN tunneling, and SCLC models. Based on these models, we simulated the current density under the electric field with various tunneling barrier (Al_2O_3) thickness layers. We varied tunneling barrier thickness from 20 Å to 40 Å, and the current density of them were plotted in **Fig. S8a**. As the current density in the direct tunneling exponentially decays with the oxide thickness, we expect the current density to be decreased by around 3 orders of magnitude per 1 nm Al_2O_3 thickness. These simulation results suggest that to set the device resistance to in ~M Ω regime, the Al_2O_3 tunneling barrier thickness should be chosen ~30Å. So, we have decided 30Å of Al_2O_3 tunneling barrier to make our devices in ~M Ω regime. We plotted the current density of an experimentally measured RRAM device with 30Å tunneling barrier thickness and 5 μm diameter (shown with open circle in **Fig. S8a**) showing 1M Ω resistance at 100 mV read voltage, showing great consistency with predictions based on the tunneling current calculations. Our methodology suggests that there is more room to modulate the barrier thickness depending on the target resistance and the device size.

Once the tunnel barrier thickness is fixed, we systematically optimized the thickness of SCLC conduction layer made of TiO_x . Thinner TiO_x will result in higher electric field across the SCLC layer for the same applied voltage and hence will increase the chances clustering of oxygen vacancies through drift to form filaments. To observe this, we chose two different TiO_x

thicknesses (6.5 nm for S3 and 40 nm for S4) leading to a difference in the applied electric field across the SCLC layer. We fitted the J-V data of both S3 and S4 using tunneling and SCLC conduction models, and we extracted the electric field applied across the TiO_x switching layer shown in **Fig. S8b** below. Since S3 has a thinner sputtered layer than S4, it causes a higher electric field across the TiO_x switching layer. The electroforming step in filamentary RRAM devices corresponds to the controlled soft-breakdown process in a thin insulator film.^{18,19} Therefore, the thin oxide film or high electric field are known to induce facile filament formation behavior in the RRAM devices. Due to high electric field in S3, it causes filament forming under a high voltage regime ($|V| > 1V$) as seen in Fig. 1g in the manuscript because the V_O filaments protrude the whole switching layer. This filamentary switching mechanism is the same as the ALD bilayer filamentary RRAM devices (S1 and S2). In a lower voltage regime ($|V| < 1V$) where electric field is not large enough to drift and cluster V_O, it shows bulk switching behavior through modulation of V_O distribution across the switching layer. In S4, however, we successfully suppressed the filament formation by reducing the electric field with a thick and amorphous TiO_x layer so that we could achieve stable bulk switching behavior.”

We also added below explanations to the “**Discussion**” section of manuscript:

“The thickness of the Al₂O₃ tunnel barrier was chosen as ~3nm to set the device resistance to ~MΩ regime (**Supplementary Note 3, Fig. S8a**). Thick TiO_x layer was needed to reduce the electric field across the V_O-rich SCLC layer so that facile filament formation due to drift and clustering of V_O could be prevented (**Fig. S8b**).”

Comment 4.

Further validation of the electrical characterization proposed, based on the two models should be carried out at various temperatures to fully validate the different temperatures (low and high) dependence of the two supposed mechanisms. If this is confirmed as trends and physical dependences, the conduction mechanism proposed would be greatly confirmed. It may also happen that the multilevel operation is affected by temperature increase and this would be very interesting experiment.

Response:

We thank the reviewer for this great suggestion to improve our analysis on the conduction mechanism.

Following the reviewer's suggestion, we carried out electrical characterization at different temperatures and validated the model. Here we explain it in detail. Our bulk switching RRAM consists of a tunnel barrier Al₂O₃ layer and a space charge limited conduction TiO_x layer. They are connected in series, so they follow the current and voltage equations below.

$$J_{tot} = J_{Al_2O_3} = J_{TiO_x}, V_{tot} = V_{Al_2O_3} + V_{TiO_x}$$

Through the Al₂O₃ tunnel barrier, direct tunneling or FN tunneling can occur depending on the voltage applied and electric field across the Al₂O₃.^{20,21} The current conduction through the TiO_x SCLC layer occurs through carrier drift and SCLC with different power law dependencies on voltage Ohm's law ($J \propto V$) and Mark-Helfrich's law ($J \propto V^{m+1}$), respectively.²²⁻
²⁴ The equations for all these mechanisms are as follows.:

$$J_{Al_2O_3} = J_{DT} + J_{FN}$$

$$J_{DT} = \left(\frac{e^2}{sh^2} \right) [m^*(\varphi_1 + \varphi_2)]^{\frac{1}{2}} V \times \exp\left(-\left(\frac{4\pi s}{h}\right) m^{*\frac{1}{2}} (\varphi_1 + \varphi_2)^{\frac{1}{2}}\right)$$

$$J_{FN} = \left(\frac{e^3 V^2}{8\pi h \Phi} \right) \exp\left[\frac{-8\pi(2m^*)^{\frac{1}{2}} \Phi^{\frac{3}{2}}}{3heV}\right]$$

$$J_{TiO_x} = J_{Drift} + J_{SCLC}$$

$$J_{Drift} = en\mu V, n \propto \exp\left(-\frac{1}{kT}\right)$$

$$J_{SCLC} = Ne\mu \left(\frac{\epsilon_s \epsilon_0}{eN_t} \right)^m \left(\frac{m}{m+1} \right)^m \left(\frac{2m+1}{m+1} \right)^{m+1} \frac{V^{m+1}}{d^{2m+1}}, m = T_c/T$$

Where e is the electron charge, s is the Al₂O₃ layer thickness, h is the Planck constant, m* is electron effective mass, φ_1 and φ_2 are barrier heights of Al₂O₃ layer from metal and TiO_x sides respectively, Φ is average barrier height of Al₂O₃ layer, n is the free electron density, μ is the mobility of electron, N_c is the effective density of state in the conduction band, ϵ_s is the dielectric constant of TiO_x, ϵ_0 is the permittivity of vacuum, N_t is the trap density, d is the TiO_x layer thickness, T_c is the characteristic temperature, m is T_c/T. T_c is the characteristic temperature defining the slope of the exponential trap distribution over bandgap energy.²²

We performed temperature-dependent I-V measurements using our bulk RRAM devices from 300K to 350K. Then, we fitted experimentally measured I-V data to the above models using MATLAB. Fig. S2 shows the experimentally measured I-V characteristics measured from 0V to 1V and fitted curves for different temperatures ranging from 300K to 350K. Our model fitting is in great agreement with experimentally measured I-V characteristics

at all temperatures and across the entire voltage range. Fig. S2b zooms in the low voltage regime (from 0V to 0.1V) where the direct tunneling across the Al₂O₃ layer dominates. Normally direct tunneling across the Al₂O₃ layer does not depend on the temperature, however, the electron drift current in TiO_x layer increases as the temperature increases due to the increase in thermal carrier density in the layer. That results in an increase in the total current as the stage temperature increases as shown in Fig. S2a. Fig. S2c focuses on the FN tunneling regime (0.25 V to 1V). At higher voltages, the band bending of the Al₂O₃ layer increases further, resulting in a transition from direct tunneling to FN tunneling across the Al₂O₃ layer. At higher voltage ranges, SCLC current also dominates the conduction across the TiO_x layer. In the high voltage regime, the current increases with temperature as it increases in low voltage regime too. However, the power dependency on voltage ($J_{\text{SCLC}} \propto V^{m+1}$, $m = T_c/T$) gets weaker as the temperature increase, resulting in smaller difference in the current for high voltages. Overall, our model shows great agreement with the I-V measurement results at all temperatures.

Finally, we thank the reviewer's suggestion about potential further research on multi-level operation under different temperatures. The device switching behavior under different temperatures would be a great research subject and as far as we know, it has never been studied for other RRAM devices in the literature as well. However, we think that it would be beyond the scope of this manuscript. We will indeed consider it for our follow-up work.

Figure S2. Experimentally measured I-V and fitted curve with different temperature from 300K to 350K. **a.** Whole range, log J – log V curves. **b.** Magnified image of $\log(J/V^2)$ vs. $1/V$ curves from 0V to 0.1V. **c.** Magnified image of $\log(J/V^2)$ vs. $1/V$ curves from 0.25V to 1V. Experimentally measured data (dotted circle) are fitted with our electrical conduction model (line).

In response to this comment, we revised our model with below explanations to the “**Characterization of bulk RRAM switching behavior**” section in the manuscript:

“We developed a model explaining current conduction in our bulk RRAM devices (**Supplementary note 1**) and fitted to experimental results shown in **Fig. 3f, g**. To further validate our model, we performed temperature-dependent I-V measurements on our bulk RRAM devices and fitted the measurement results to our model (**Fig. S2**). Our model based on direct tunneling, Fowler-Nordheim tunneling, and SCLC shows great agreement with the current voltage characteristics at all temperatures and voltage ranges.”

Supplementary note 1

“Our bulk switching RRAM consists of a tunnel barrier Al₂O₃ layer and a space charge limited conduction TiO_x layer. They are connected in series, so they follow the current and voltage equations below.

$$J_{tot} = J_{Al_2O_3} = J_{TiO_x}, V_{tot} = V_{Al_2O_3} + V_{TiO_x} \quad (S1)$$

Through the Al₂O₃ tunnel barrier, direct tunneling or FN tunneling can occur depending on the voltage applied and electric field across the Al₂O₃.^{20,21} The current conduction through the TiO_x SCLC layer occurs through carrier drift and SCLC with different power law dependencies on voltage Ohm’s law ($J \propto V$) and Mark-Helfrich’s law ($J \propto V^{m+1}$), respectively.²²⁻
²⁴ The equations for all these mechanisms are as follows.:

$$J_{Al_2O_3} = J_{DT} + J_{FN} \quad (S2)$$

$$J_{DT} = \left(\frac{e^2}{sh^2}\right) [m^*(\varphi_1 + \varphi_2)]^{\frac{1}{2}} V \times \exp\left(-\left(\frac{4\pi s}{h}\right) m^{*\frac{1}{2}}(\varphi_1 + \varphi_2)^{\frac{1}{2}}\right) \quad (S3)$$

$$J_{FN} = \left(\frac{e^3 V^2}{8\pi h \Phi}\right) \exp\left[\frac{-8\pi(2m^*)^{\frac{1}{2}} \Phi^{\frac{3}{2}}}{3heV}\right] \quad (S4)$$

$$J_{TiO_x} = J_{Drift} + J_{SCLC} \quad (S5)$$

$$J_{Drift} = en\mu V, n \propto \exp\left(-\frac{1}{kT}\right) \quad (S6)$$

$$J_{SCLC} = Ne\mu \left(\frac{\varepsilon_s \varepsilon_0}{eN_t}\right)^m \left(\frac{m}{m+1}\right)^m \left(\frac{2m+1}{m+1}\right)^{m+1} \frac{V^{m+1}}{d^{2m+1}}, m = T_c/T \quad (S7)$$

Where e is the electron charge, s is the Al₂O₃ layer thickness, h is the Planck constant, m* is electron effective mass, φ_1 and φ_2 are barrier heights of Al₂O₃ layer from metal and TiO_x sides respectively, Φ is average barrier height of Al₂O₃ layer, n is the free electron density, μ is the mobility of electron, N_c is the effective density of state in the conduction band, ε_s is the dielectric constant of TiO_x, ε_0 is the permittivity of vacuum, N_t is the trap density, d is the TiO_x

layer thickness, T_c is the characteristic temperature, m is T_c/T . T_c is the characteristic temperature defining the slope of the exponential trap distribution over bandgap energy.²²

Comment 5.

The tested devices have still large size; DC I-V switching curves of trilayer bulk RRAM is tested in cells from 3 μm to 10 μm . What about maintaining the observed performance and properties when scaling the size of the cells? Any foreseen barriers involving for instance higher variability of characteristics due to cell scaling?

Response:

We thank the reviewer for pointing this out.

The root cause of the higher variability of characteristics due to cell scaling is related to the variability in the number of charged carriers or particles for the memory devices. For instance, the charge trap memory with 10-nm technology node can store only 10 electrons per device which causes severe variability issues in the device characteristics.²⁵ Therefore, we calculated the number of defects in our bulk RRAM devices based on the stoichiometry of the film. We estimated the stoichiometry of the sputtered TiO_x layer as $\text{TiO}_{1.85}$, which corresponds to 4.77×10^{21} V_O defects/ cm^3 in the layer. When we assume the device scales down to 20 \times 20 nm^2 cell sizes with 40 nm thickness, ~ 76000 V_O defects exist in a single device. The order of V_O defects in a single cell is 4 orders higher than the number of trapped electrons in a 10nm tech node flash memory devices, so variability related to the low number of defects is not expected to be a major problem for our bulk RRAM devices.

In addition, the high variability in the conventional RRAM devices come from the stochastic nature of the filaments. Since both set and reset processes are determined by the stochastic movement of atoms in the switching oxides, the device-to-device and cycle-to-cycle variability have been a main problem of filamentary RRAM devices. In our bulk RRAM devices, resistance is modulated by controlling the defect concentration in the switching layer so that the variability problem of stochastic 1D filaments can be resolved. In addition, the previous study about bulk RRAM devices demonstrated uniform resistive switching characteristics when the device is scaled down to 60 \times 60 nm^2 .²⁶

Finally, for the scaled bulk RRAM devices, the tunnel oxide and the TiO_x switching layer thicknesses need to be reoptimized to match $\text{M}\Omega$ resistance level. Since our bulk RRAM device shows area-scaling effects (**Fig. 3b**), the device resistance would be increased when it is in nm scale. We estimated that the device resistance can be tuned to $\text{M}\Omega$ level by changing

the Al₂O₃ barrier thickness as ~ 2nm, for the nm scale device demonstration. In response to this comment, we added the below explanations in the “**Discussion**” section of the manuscript:

“Next step for the bulk RRAM technology is scaling device dimensions to the nm regime. One potential concern could be variability for nanoscale devices. For memory technologies, the root cause of the higher variability of characteristics due to cell scaling is related to the number of charged carriers or particles. For instance, the charge trap memory with 10-nm technology node can store only 10 electrons per device which cause severe variability issues in the device characteristics.²⁵ Therefore, we calculated the number of defects in our bulk RRAM devices based on the stoichiometry of the film. We estimated the stoichiometry of the sputtered TiO_x layer as TiO_{1.85}, which corresponds to 4.77×10^{21} V_O defects/cm³ in the layer. When we assume the device scales down to 20 nm size with 40 nm thickness, ~76000 V_O defects exist in a single device. The order of V_O defects in a single cell is 4 orders higher than the number of trapped electrons in a 10nm tech node flash memory devices, so variability related to the low number of defects is not expected to be a major problem for our bulk RRAM devices. In addition, the high variability in the conventional RRAM devices come from the stochastic nature of the filaments. Since both set and reset processes are determined by the stochastic movement of atoms in the switching oxides, the device-to-device and cycle-to-cycle variability have been a main problem of filamentary RRAM devices. In our bulk RRAM devices, resistance is modulated by controlling the defect concentration in the switching layer so that the variability problem of stochastic 1D filaments can be resolved. In addition, the previous study about bulk RRAM devices demonstrated uniform resistive switching characteristics when the device is scaled down to 60×60 nm².²⁶ Finally, for the scaled bulk RRAM devices, the tunnel oxide and the TiO_x switching layer thicknesses could be reoptimized to match MΩ resistance level.”

Comment 6.

While I find the validation of the hardware implementation of SNN for a navigation/racing task with 16 x 16 cross bars for weight mapping very interesting and a great results, I am not sure how at architecture level this will compare with other technologies implementing the same. A short discussion with some clear benchmarks of gains would clarify the major advantages beyond the functional aspect.

Response: We thank the reviewer for this encouraging comment.

We have already mentioned, the several advantages of this bulk switching RRAM technology from the pure device technology point of view in Table S2. To compare our work to other technologies at the architecture level, we simulated the SNN trained for the same navigation tasks while implementing the required synaptic weights using different RRAM technologies. First, we mapped all synaptic weights on conductance levels of different RRAM technologies and implemented the weights in the simulation framework running the navigation task on different racetrack maps. For every racetrack map, we obtained the total energy consumption required to perform MVM operations using the RRAM arrays based on different RRAM technologies while performing the navigation task. We analyzed the total energy needed to complete each racetrack map. Energy consumed by peripheral circuits cannot be included in energy computation since we do not have access to peripheral designs for the other technologies and it would depend on the design and many other things such as technology node, etc. So, our energy computation reflects the energy consumed by RRAM synaptic cores.

Figure S6 shows the average energy consumed during navigation by including all the racetracks for different RRAM technologies in comparison to our bulk RRAM. **Figure S7** shows the total energy consumed for navigating three representative racetrack maps (i.e., Catalunya, Nuerburgring, and Oschersleben) based on synaptic weight implementation using different RRAM technologies. Both results show that our trilayer bulk RRAM can substantially reduce energy consumption for MVM operations of our SNN model trained for navigation.

Figure S6. Average energy consumption and its variance during navigation of all 15 racetracks by using different RRAM technologies for implementing synaptic weights.

Figure S7. Total energy consumption during navigation of three representative racetracks by using different RRAM technologies for implementing synaptic weights.

Now, we have included these results in the supplementary file as **Figure S6** and **Figure S7**, respectively.

We have also added the following text in the manuscript section **“Hardware SNN implementation with RRAM crossbars.”**.

“Furthermore, to compare our work to other technologies (i.e., HfO_x:Si¹⁵, HfO_x/TaO_y¹⁴, and Al₂O₃/TiO₂¹³) at the architecture level, we simulated the energy consumption by the SNN model trained for the same navigation tasks while implementing the required synaptic weights using these different RRAM technologies. **Fig. S6** and **S7** show total energy needed to navigate all racetracks and individual racetracks, respectively. Our results suggest that our trilayer bulk RRAM substantially (more than order of two) reduce energy consumed by the synaptic arrays in comparisons to other RRAM technologies.”

Reviewer #3 (Remarks to the Author):

The authors report non-filamentary synaptic devices using a trilayer oxide structure to overcome limitations of filamentary memristor devices reported in literature. Using multiple layers including one that is insulating but thin, they argue that filamentary switching can be avoided and show some data for area scaling to support this hypothesis. Comments:

Comment 1.

Please explain the vacancy concentration in the TiO₂ layer and what values are expected in their samples and their distribution to support the claims concerning the conduction mechanism.

Response:

We thank the reviewer for this insightful comment. We have estimated the stoichiometry of the film by two different methods. First, we deposited sputtered TiO_x layer with 5% O₂/Ar ratio during the sputtering process. In a previous study, the sputtered film with 5% O₂/Ar ratio showed the stoichiometry of TiO_{1.87±0.02} by X-ray photoelectron spectroscopy method.²⁷ Second, we have interpolated the measured conductivity value to a conductivity vs. oxygen deficiency relation in the titanium oxide system⁴. The sheet resistance of 50nm-thick ALD TiO₂ and sputtered TiO_x layers were measured by the 4-point probe method, and conductivity was calculated. Based on our conductivity measurements, the stoichiometry of sputtered TiO_x layer and ALD TiO₂ layer are expected to be TiO_{1.85} and TiO_{1.96}, respectively. We intentionally introduce the V_O in the sputtered TiO_x layer so that we can move the defects back and forth by electrical stimuli. In terms of the conduction mechanism through the switching layer, SCLC conduction occurs through the layer in comparatively higher electric field than where the ohmic conduction occurs in the TiO_x layer. Following the reviewer's comment, we added the following explanations to the supplementary information:

“We have estimated the stoichiometry of the sputtered TiO_x and ALD TiO₂ film in **Table S1**. We expect the stoichiometry of the sputtered TiO_x layer to be around TiO_{1.85} based on our experimentally measure conductivity and O₂/Ar ratio during the sputtering process^{4,27}. Due to the intentionally induced V_O defects, SCLC conduction occurs through the layer in comparatively higher electric field than where the ohmic conduction occurs in the TiO_x layer.”

	Sheet Resistance (Ω/□)	Conductivity (Ω·m) ⁻¹	Estimation by Rs	Estimation by O ₂ /Ar ratio
ALD TiO ₂	125M	0.16	TiO _{1.96}	N/A
Sputtered TiO _x	1.48M	13.5	TiO _{1.85}	TiO _{1.87±0.02}

Table S1. Stoichiometry estimation by conductivity measurement and XPS analysis.

Comment 2.

Non-filamentary synaptic updates in oxides have indeed been previously reported (Nature Comms, 11, 2245, 2020), please compare and contrast to present results to place work in context of literature.

Response:

We thank the reviewer for the kind suggestion. Following the reviewer's suggestion, we prepared a benchmark Table S2 comparing previous filamentary and bulk RRAM technologies to our device for neuromorphic applications. We have added the non-filamentary synaptic memory using hydrogenated NdNiO₃ to the benchmark table.¹⁷ (**Table S2**) NdNiO₃-based devices can have many states by changing the hydrogen cloud near the hydrogen-active Pt electrode using sub-microsecond voltage pulses. In that reference, H-T. Zhang et al. demonstrated 300µm diameter devices operating in the kΩ regime. It is expected that the resistance can be increased to MΩ regime when the devices are scaled down to a few micrometer cells, which is a promising next step for NdNiO₃ MΩ level operation needed for a large array-scale demonstration. NdNiO₃ needs high process temperatures (T > 500°C) so further process development to enable fabrication at low temperatures (T < 400°C) will be necessary for CMOS compatibility. Our bulk RRAM devices based on Al₂O₃/TiO₂ materials not only address the limitations of filamentary RRAM devices but are also compatible with integration at the CMOS BEOL to enable a high-density 3D compute-in-memory platform for neuromorphic applications.

Following the reviewer's comment, we added the following explanations to the **supplementary note 4**:

“Right three columns compare recently developed bulk RRAM devices based on NdNiO₃, Pr_{0.7}Ca_{0.3}MnO₃ and Al₂O₃/TiO₂/TiO_x to address the limitations of filamentary RRAM devices explained above.^{16,17} First, these bulk RRAM technologies do not need any initial high voltage forming step. The resistance modulation occurs in MΩ regime ensuring stable and reliable large size array operations. The cell size of these bulk RRAM is larger than the filamentary RRAM since the technology is less mature and has not yet translated into foundries in contrast to HfO_x based filamentary RRAM. Since all bulk RRAM devices exhibit higher resistance than filamentary RRAM, the energy consumption is much lower for the read operation. In addition, many number of states could be achieved to represent the synaptic

weights in neuromorphic computing applications. Among the three bulk RRAM technologies, only our work presents a large array-scale demonstration. Furthermore, perovskite-based bulk RRAM technologies need high process temperature ($T > 500^{\circ}\text{C}$) so further process development to enable fabrication at low temperatures ($T < 400^{\circ}\text{C}$) will be necessary for CMOS BEOL compatibility. Our bulk RRAM devices based on $\text{Al}_2\text{O}_3/\text{TiO}_2$ materials not only address the limitations of the mature filament RRAM technologies but are also compatible with integration at the CMOS BEOL to enable a high-density 3D compute-in-memory platform for neuromorphic applications. Based on **Table S2**, the key figures of merit of our bulk RRAM technology RRAM technology can be summarized as follows; forming-free operation, CMOS BEOL compatibility, high R_{on} and R_{off} that enables reliable read and write in large scale crossbar arrays and low energy operation, low switching voltages, high number of conductance states, endurance comparable to RRAM technologies, and much lower total read energy.”

Table S2. Benchmarking of conventional RRAM technologies for neuromorphic applications.

RRAM Technologies	$\text{Al}_2\text{O}_3/\text{TiO}_2$ [13]	$\text{HfO}_2/\text{TaO}_x$ [14]	$\text{HfO}_2:\text{Si}$ [15]	PCMO [16]	NdNiO_3 [17]	$\text{Al}_2\text{O}_3/\text{TiO}_2/\text{TiO}_x$ [This work]
Switching Type	Filamentary	Filamentary	Filamentary	Bulk	Bulk	Bulk
Forming voltage	~3.3V	~4.0V	~2.0V	None	None	None
CMOS Compatibility	✓	✓	✓	X	X	✓
R_{on}	16 $\text{K}\Omega$	23.8 $\text{K}\Omega$	6 $\text{K}\Omega$	6.25 $\text{M}\Omega$	2 $\text{K}\Omega$	2.78 $\text{M}\Omega$
R_{off}	333 $\text{K}\Omega$	500 $\text{K}\Omega$	600 $\text{K}\Omega$	157 $\text{M}\Omega$	4.5 $\text{K}\Omega$	6.67 $\text{M}\Omega$
Cell area (μm^2)	0.0625 μm^2	0.360 μm^2	0.04 μm^2	0.0225 μm^2	70700 μm^2	19.6 μm^2
Switching voltage	1.2 V / -1.4 V	+1.6 V / -1.9 V	+1.7 V / -2.0 V	-3.0V / +3.0V	-3.0 V / 5.4 V	-1.8V / + 0.65V
Pulse width	1ms	10us	100ns	1ms	960ns	5ms
# of states	16	8	2	50	32	100
Endurance	10^6	10^6	10^5	10^4	-	2^*10^5
Total Energy per read operations	51.0 fJ	44.02 fJ	167.7 fJ	1.04 fJ	722.2 fJ	~ 0.51 fJ

Comment 3.

It is not clear the difference between S3 and S4 samples why there is a different behavior. What is it about the TiO_x layer that causes this?

Response:

We thank the reviewer for pointing this out. Before we explain the difference between S3 and S4, we should clarify the mechanism difference between filamentary RRAM and bulk RRAM devices. Please refer to comment 4.

The difference between S3 and S4 is the applied electric field across the sputtered TiO_x layer. We fitted the J-V data of both S3 and S4 using tunneling and SCLC conduction models, and we extracted the electric field applied across the TiO_x switching layer (**Fig. S8b**). Since S3 has a thinner sputtered layer than S4, it causes a higher electric field across the TiO_x switching layer. The electroforming step in filamentary RRAM devices corresponds to the controlled soft-breakdown process in a thin insulator film.^{18,19} Therefore, the thin oxide film or high electric field are known to induce facile filament formation behavior in the RRAM devices. Due to the high electric field in S3, it causes filament forming under a high voltage regime ($|V| > 1V$) (**Fig. 1g** in the manuscript) because the V_O filaments protrude the whole switching layer. This filament switching mechanism is the same as the ALD bilayer filamentary RRAM devices (S1 and S2). In a lower voltage regime ($|V| < 1V$), it shows bulk switching behavior due to a change in defect distribution across the switching layer. In S4, however, we successfully suppressed the filament formation with a thick and amorphous TiO_x layer so that we could achieve stable bulk switching behavior.

Following the reviewer's comment, we added the following explanations to the **supplementary note 3**:

“The difference between S3 and S4 is the applied electric field across the sputtered TiO_x layer. We fitted the J-V data of both S3 and S4 using tunneling and SCLC conduction models, and we extracted the electric field applied across the TiO_x switching layer (**Fig. S8b**). Since S3 has a thinner sputtered layer than S4, it causes a higher electric field across the TiO_x switching layer. The electroforming step in filamentary RRAM devices corresponds to the controlled soft-breakdown process in a thin insulator film.^{18,19} Therefore, the thin oxide film or high electric field are known to induce facile filament formation behavior in the RRAM devices. Due to high electric field in S3, it causes filament forming under a high voltage regime ($|V| > 1V$) because the V_O filaments protrude the whole switching layer. This filament switching mechanism is the same as the ALD bilayer filamentary RRAM devices (S1 and S2). In a lower voltage regime ($|V| < 1V$), it shows bulk switching behavior due to a change in defect distribution across the switching layer. In S4, however, we successfully suppressed the filament formation with a thick and amorphous TiO_x layer so that we could achieve stable bulk switching behavior.”

Figure S8b. Electric field applied across the TiO_x sputtered layer in S3 and S4.

Comment 4.

Why is the switching polarity different in S4 compared to other samples, if the same mechanism persists but the concentration of defects is reduced according to the authors?

Response:

We thank the reviewer for pointing this out. We would like to clarify the switching polarity of all the samples first. The ALD-deposited bilayer samples (S1 and S2) exhibit filamentary switching where a positive voltage reduces resistance (set) and negative voltage increases resistance (reset). S3 shows filamentary switching with same polarity as S1 and S2 in high voltage regime, but it also exhibits bulk switching in low voltage regime. The bulk switching observed in S3 and S4 is opposite polarity to filamentary switching. In bulk switching, positive voltage increases resistance (reset) and negative voltage reduces resistance (set) without abrupt resistance change.

The opposite polarity comes from the different switching mechanism of filamentary and bulk RRAM. In the filamentary RRAM, V_O defects are well known to be mobile with external electrical and thermal stimuli.⁵ Filamentary RRAM needs an initial electroforming step which forms the V_O defect filaments between two electrodes. Once the filaments have formed, bipolar switching takes place due to the forming and rupturing of the filaments. Positive voltage to top electrode forms the filament and a negative voltage to top electrode attracts the V_O and ruptures the filament (**Fig. 1a**).

In our bulk RRAM devices, we deposited an amorphous, porous, and V_O-rich thick TiO_x layer instead of having a thick crystalline ALD TiO₂ layer. Due to the absence of fast diffusion paths or accumulation sites for V_O defects in the amorphous phase, the V_O defects are not clustered in specific locations that facilitate filament formation. The filament formation is

effectively suppressed in an amorphous layer as compared to the crystalline phase in RRAM devices.⁸ V_O defects will drift homogeneously throughout the entire area of the layer rather than forming defects-clustered filaments, following the direction of the electric field, enabling bulk switching instead of filamentary switching. When a positive voltage is applied to the top electrode, the V_O are pushed downwards following the electric field and the V_O concentration in the TiO_x layer is reduced (**Fig. 1a**). Space-charge-limited-conduction (SCLC) dominates the conduction in the TiO_x layer. Since the V_O concentration in the TiO_x layer is reduced, the SCLC current decreases confirming that the device is reset to a higher resistance state. We successfully confirmed this model by fitting IV characteristics at different resistance states to our model including tunneling and SCLC conduction (**Fig. 3g**) and extracted the trap density (**Fig. 3h**). The trap density is decreased after the device is programmed to higher resistance states using positive bias.

In response to this comment, we added explanations below in the manuscript:

“Bulk switching can be better understood by reviewing filamentary RRAM first. In the filamentary RRAM, V_O defects are well known to be mobile with external electrical and thermal stimuli.⁵ Filamentary RRAM needs an initial high voltage electroforming step which forms the V_O defect filaments between two electrodes. Once the filaments have formed, bipolar switching takes place due to the forming and rupturing of the filaments. Meanwhile, the grain boundaries are well known to be the high diffusivity paths of small ions such as oxygens or hydrogens in crystalline oxides.^{6,7} Especially for our polycrystalline filamentary RRAM devices (S1, S2), the grain boundaries play an important role in charge transport and V_O accumulation and diffusion. Due to these diffusivity paths, filament formation and rupture easily occur in the filamentary RRAM devices.

In our bulk RRAM devices, we deposited an amorphous, porous, and V_O -rich thick TiO_x layer instead of having a crystalline ALD TiO_2 layer. Due to the absence of fast diffusion paths or accumulation sites for V_O defects in the amorphous phase, the V_O defects are not clustered in specific locations that facilitate filament formation. The filament formation is effectively suppressed in an amorphous layer as compared to the crystalline phase in RRAM devices.⁸ V_O defects will drift homogeneously throughout the entire area of the layer rather than forming defects-clustered filaments, following the direction of the electric field, enabling bulk switching instead of filamentary switching. When a positive voltage is applied to the top electrode, the V_O are pushed downwards following the electric field and the V_O concentration in the TiO_x layer is reduced. Space-charge-limited-conduction (SCLC) dominates the

conduction in the TiO_x layer. Since the V_O concentration in the TiO_x layer is reduced, the SCLC current decreases confirming that the device is reset to a higher resistance state. We successfully confirmed this model by fitting IV characteristics at different resistance states to our model including tunneling and SCLC conduction (**Fig. 3g**) and extracted the trap density (**Fig. 3h**). The trap density is decreased after the device is programmed to higher resistance states using positive bias.”

Comment 5.

Have the authors tried different layer thickness of Al_2O_3 to verify the mechanism?

Response:

We thank the reviewer for pointing this out. We would like to explain how we chose the Al_2O_3 tunnel barrier thickness. As described in the manuscript, current conduction occurs through direct tunneling, FN tunneling, and SCLC models. Based on these models, we simulated the current density under the electric field with various tunneling barrier (Al_2O_3) thickness layers. We varied tunneling barrier thickness from 20 Å to 40 Å, and the current density of them were plotted in **Fig. S8a**. As the current density in the direct tunneling exponentially decays with the oxide thickness, we expect the current density to be decreased by around 3 orders of magnitude per 1 nm Al_2O_3 thickness. These simulation results suggest that to set the device resistance to in $\sim\text{M}\Omega$ regime, the Al_2O_3 tunneling barrier thickness should be chosen $\sim 30\text{\AA}$. So, we have decided 30Å of Al_2O_3 tunneling barrier to make our devices in $\sim\text{M}\Omega$ regime. We plotted the current density of an experimentally measured RRAM device with 30Å tunneling barrier thickness and 5µm diameter (shown with open circle in **Fig. S8a**) showing 1MΩ resistance at 100 mV read voltage, showing great consistency with predictions based on the tunneling current calculations. Our methodology suggests that there is more room to modulate the barrier thickness depending on the target resistance and the device size.

Figure S8a. Current density versus applied voltage across the RRAM switching layer. The Al₂O₃ thickness was varied from 20Å to 40Å to determine the experimental Al₂O₃ thickness. Open circle curve shows experimentally measured current density from an RRAM device with 30Å tunneling barrier thickness and 5µm diameter, exhibiting 1MΩ resistance at 100mV read voltage.

In response to this comment, we added explanations below in the “**Discussion**” section of manuscript:

“The thickness of the Al₂O₃ tunnel barrier was chosen as ~3nm to set the device resistance to ~MΩ regime (**Supplementary Note 3, Fig. S8a**). Thick TiO_x layer was needed to reduce the electric field across the V_O-rich SCLC layer so that facile filament formation due to drift and clustering of V_O could be prevented (**Fig. S8b**). We achieved multi-level, uniform bulk switching in MΩ regime without a compliance current in the bulk RRAM devices.”

Comment 6.

If ALD of alumina provides good quality film (ie serves as insulating barrier), why ALD of Tio2 layer results in low quality samples leading to filamentary switching? Is there some process issue or is this intentional?

Response:

We thank the reviewer for pointing this out. We deposited ALD TiO₂ layer with TiCl₄ and H₂O precursors under 250°C (~0.43nm/cycle), and we expected the stoichiometry of ALD TiO₂ layer to be around TiO_{1.96} (**Table. S1**). When the growth temperature is higher than 200°C, the high-quality anatase TiO₂ film would be formed with low impurity concentrations.²⁸ The reason for filamentary resistive switching is not related to the quality of the oxide layer but the crystallinity of the film. As shown in **Fig. S1**, the ALD TiO₂ layer has an anatase crystalline phase with grain boundaries. The grain boundaries are well known to be the high diffusivity paths of small ions such as oxygens or hydrogens in crystalline oxides.^{6,7} Especially for our polycrystalline filamentary RRAM devices (S1, S2), the grain boundaries play an important role in charge transport and V_O accumulation and diffusion. Due to these diffusivity paths, filament formation and rupture easily occur in the filamentary RRAM devices.

Comment 7.

The authors use porous films but clearly they are not controlling the pore location or such. In this scenario, how can the samples be considered reproducible and overcome the variability limitations mentioned in the introduction? Please clarify.

Response:

We thank the reviewer for pointing this out.

We achieved highly uniform switching characteristics across the wafer, showing low cycle-to-cycle (C2C) and device-to device (D2D) variability for the bulk RRAM devices. In **Fig. 1d**, we have shown almost perfectly overlapped C2C DC switching characteristics in S4. **Fig. 3c** shows uniform cumulative distribution function (CDF) of bulk RRAM pristine device resistances in different size cells. **Fig. 3d** shows uniform CDF for low and high resistance states. We have also shown the uniform resistance switching using voltage (**Fig. 4, Fig. S2**). Overall, extensive characterization results suggest that variability is not a concern for our bulk RRAM devices. To further address the reviewer's concern about the film uniformity in the sputtered layer, we deposited 50-nm thick of sputtered TiO_x layer on a 4-inch SiO₂/Si wafer and measured the sheet resistance across the wafer. We confirmed highly uniform sheet resistance and device switching behavior across the whole wafer which can address the reproducibility and variability issues in filamentary RRAM devices.

Following the reviewer's comment, we added the following explanations to the supplementary information:

“To investigate uniformity of the films, we deposited 50-nm thick of sputtered TiO_x layer on a 4-inch SiO₂/Si wafer and measured the sheet resistance across the wafer. We confirmed highly uniform sheet resistance and device switching behavior across the whole wafer which can address the reproducibility and variability issues in filamentary RRAM devices.”

	Sheet Resistance (Ω/\square)
Center	1.48M
Middle	1.53M
Edge	1.53M

Table S3. Sheet resistance uniformity of sputtered TiO_x layer across the wafer.

- 1 Jacob, K. & Gupta, S. Calciothermic reduction of TiO₂: A diagrammatic assessment of the thermodynamic limit of deoxidation. *JOM* **61**, 56-59 (2009).
- 2 Kang, M. & Son, J. Off-state current reduction in NbO₂-based selector device by using TiO₂ tunneling barrier as an oxygen scavenger. *Applied Physics Letters* **109** (2016).

- 3 Calka, P. *et al.* Engineering of the Chemical Reactivity of the Ti/HfO₂ Interface for RRAM: Experiment and Theory. *ACS applied materials & interfaces* **6**, 5056-5060 (2014).
- 4 Hoskins, B. D. & Strukov, D. B. Maximizing stoichiometry control in reactive sputter deposition of TiO₂. *Journal of Vacuum Science & Technology A* **35** (2017).
- 5 Waser, R. & Aono, M. Nanoionics-based resistive switching memories. *Nature materials* **6**, 833-840 (2007).
- 6 Park, J., Yoon, H., Sim, H., Choi, S.-Y. & Son, J. Accelerated hydrogen diffusion and surface exchange by domain boundaries in epitaxial VO₂ thin films. *ACS nano* **14**, 2533-2541 (2020).
- 7 Navickas, E. *et al.* Fast oxygen exchange and diffusion kinetics of grain boundaries in Sr-doped LaMnO₃ thin films. *Physical Chemistry Chemical Physics* **17**, 7659-7669 (2015).
- 8 Grossi, A. *et al.* Electrical characterization and modeling of 1T-1R RRAM arrays with amorphous and poly-crystalline HfO₂. *Solid-State Electronics* **128**, 187-193 (2017).
- 9 Yu, S. Neuro-inspired computing with emerging nonvolatile memories. *Proceedings of the IEEE* **106**, 260-285 (2018).
- 10 Shi, Y. *et al.* Adaptive quantization as a device-algorithm co-design approach to improve the performance of in-memory unsupervised learning with SNNs. *IEEE Transactions on Electron Devices* **66**, 1722-1728 (2019).
- 11 Gholami, A. *et al.* in *Low-Power Computer Vision* 291-326 (Chapman and Hall/CRC, 2022).
- 12 Nagel, M., Baalen, M. v., Blankevoort, T. & Welling, M. in *Proceedings of the IEEE/CVF International Conference on Computer Vision*. 1325-1334.
- 13 Kim, H., Mahmoodi, M., Nili, H. & Strukov, D. B. 4K-memristor analog-grade passive crossbar circuit. *Nature communications* **12**, 5198 (2021).
- 14 Wan, W. *et al.* A compute-in-memory chip based on resistive random-access memory. *Nature* **608**, 504-512 (2022).
- 15 Grenouillet, L. *et al.* in *2021 IEEE International Memory Workshop (IMW)*. 1-4 (IEEE).
- 16 Moon, K. *et al.* in *2015 IEEE International Electron Devices Meeting (IEDM)*. 17.16. 11-17.16. 14 (IEEE).
- 17 Zhang, H.-T. *et al.* Perovskite neural trees. *Nature communications* **11**, 2245 (2020).
- 18 Park, T. H. *et al.* Thickness-dependent electroforming behavior of ultra-thin Ta₂O₅ resistance switching layer. *physica status solidi (RRL)–Rapid Research Letters* **9**, 362-365 (2015).
- 19 Hu, R. *et al.* Investigation of Resistive Switching Mechanisms in Ti/TiO_x/Pd-Based RRAM Devices. *Advanced Electronic Materials* **8**, 2100827 (2022).
- 20 Lenzlinger, M. & Snow, E. Fowler-Nordheim tunneling into thermally grown SiO₂. *Journal of Applied physics* **40**, 278-283 (1969).
- 21 Simmons, J. G. Electric tunnel effect between dissimilar electrodes separated by a thin insulating film. *Journal of applied physics* **34**, 2581-2590 (1963).
- 22 Mark, P. & Helfrich, W. Space-charge-limited currents in organic crystals. *Journal of Applied Physics* **33**, 205-215 (1962).
- 23 Lampert, M. A. Simplified theory of space-charge-limited currents in an insulator with traps. *Physical Review* **103**, 1648 (1956).
- 24 Lampert, M. A. & Schilling, R. B. in *Semiconductors and semimetals* Vol. 6 1-96 (Elsevier, 1970).

- 25 Nayfeh, A. & El-Atab, N. *Nanomaterials-Based Charge Trapping Memory Devices*. (Elsevier, 2020).
- 26 Zhang, H. *et al.* Understanding the coexistence of two bipolar resistive switching modes with opposite polarity in Pt/TiO₂/Ti/Pt nanosized ReRAM devices. *ACS applied materials & interfaces* **10**, 29766-29778 (2018).
- 27 El Mesoudy, A. *et al.* Band gap narrowing induced by oxygen vacancies in reactively sputtered TiO₂ thin films. *Thin Solid Films* **769**, 139737 (2023).
- 28 Ylivaara, O. M. *et al.* Mechanical and optical properties of as-grown and thermally annealed titanium dioxide from titanium tetrachloride and water by atomic layer deposition. *Thin Solid Films* **732**, 138758 (2021).

REVIEWERS' COMMENTS

Reviewer #1 (Remarks to the Author):

The authors addressed all my comments. I have no further comment on the authors' manuscript. Now this manuscript can be published in Nature Comm.

Reviewer #2 (Remarks to the Author):

I find the answers provided by the authors very comprehensive and to the point. Particularly, the Table S2 with benchmarking of conventional RRAM technologies for neuromorphic applications is very helpful and position the work with respect to the state of the art.

Additionally, the new discussion about the approach to find the "magic" combination of tri-layers with the thicknesses proposed for sample S4 and the temperature dependence are clarifying all my doubts and adding significant value to the paper findings. The authors did a great job!

All in all, I am very satisfied with both the answers and the new version of the manuscript that is suitable for publication as is.

Reviewer #3 (Remarks to the Author):

I very much appreciate the authors technical responses to the comments from the first round. In my opinion, the authors have attempted reasonably to respond to most of the comments. there is however one technical issue that results in internal inconsistency in the manuscript. This relates to the presence of opposite polarity switching in Sample S4 compared to their rest of samples. The authors should perform some basic characterization of S4 and perhaps contrast against a sample that shows opposite behavior such as S3 to demonstrate this is in fact a real result and not an artifact. Since almost all the device results are interpreted in this paper in terms of non-filamentary transport and this is pretty much the differentiator or 'novelty' aspect of the paper, it is important for the authors to present a mechanistic data driven analysis. Otherwise, this aspect can weaken the rest of the device discussion / results and reasoning for the non-filamentary conduction results in the paper.

Reviewer #1 (Remarks to the Author):

The authors addressed all my comments. I have no further comment on the authors' manuscript. Now this manuscript can be published in Nature Comm.

Response:

We thank the reviewer for his/her insightful comments and suggestions that greatly helped to improve our manuscript. We are glad to hear that we addressed all the comments from Reviewer 1.

Reviewer #2 (Remarks to the Author):

I find the answers provided by the authors very comprehensive and to the point. Particularly, the Table S2 with benchmarking of conventional RRAM technologies for neuromorphic applications is very helpful and position the work with respect to the state of the art.

Additionally, the new discussion about the approach to find the "magic" combination of tri-layers with the thicknesses proposed for sample S4 and the temperature dependence are clarifying all my doubts and adding significant value to the paper findings. The authors did a great job! All in all, I am very satisfied with both the answers and the new version of the manuscript that is suitable for publication as is.

Response:

We thank the reviewer for his/her insightful comments and suggestions that greatly helped to improve our manuscript. We are glad to hear that we addressed all the comments from Reviewer 2.

Reviewer #3 (Remarks to the Author):

I very much appreciate the authors technical responses to the comments from the first round. In my opinion, the authors have attempted reasonably to respond to most of the comments. there is however one technical issue that results in internal inconsistency in the manuscript. This relates to the presence of opposite polarity switching in Sample S4 compared to their rest of samples. The authors should perform some basic characterization of S4 and perhaps contrast against a sample that shows opposite behavior such as S3 to demonstrate this is in fact a real result and not an artifact. Since almost all the device results are interpreted in this paper in terms of non-filamentary transport and this is pretty much the differentiator or ‘novelty’ aspect of the paper, it is important for the authors to present a mechanistic data driven analysis. Otherwise, this aspect can weaken the rest of the device discussion / results and reasoning for the non-filamentary conduction results in the paper.

Response:

We thank the reviewer for this comment. I think that reviewer 3 was confused about the polarity of switching between S3 and S4. S3 does not show opposite polarity switching to S4. There is no inconsistency in the polarity of bulk and filamentary switching. S3 exhibits both filamentary and bulk switching, for low bias it exhibits bulk switching but as the biased is increased a filament is formed and from that point on filamentary switching dominates. As seen below in Fig 1 g and h, S3 and S4 exhibit bulk switching with the same exact polarity (red curves): positive voltage increases resistance (reset) and negative voltage reduces resistance (set) without abrupt resistance change. The red bulk switching shows the same polarity, as indicated by the red arrows. So, all devices showing bulk switching devices exhibit positive reset and negative set, independent of their oxide thicknesses.

Maybe the reviewer was confused because we did not have black up arrows for filamentary switching for S3. We now added two more black arrows to Fig. 1 g to clearly point out filamentary switching in S3. We also added below text to the caption.

“Black arrows show the polarity of filamentary switching, while red arrows show polarity of bulk switching.”

All filamentary devices exhibit opposite polarity switching where a positive voltage reduces resistance (set) and negative voltage increases resistance (reset) (Fig 1 e, f below). Again, all filamentary devices exhibit positive set, negative reset consistently for S1, S2 and S3. It important to S3 exhibits both filamentary (black curves) and bulk switching(red curves).

The opposite polarity comes from the different switching mechanism of filamentary and bulk RRAM. In the filamentary RRAM, V_O defects are well known to be mobile with external electrical and thermal stimuli.¹ Filamentary RRAM needs an initial electroforming step which forms the V_O defect filaments between two electrodes. Once the filaments have formed, bipolar switching takes place due to the forming and rupturing of the filaments. Positive voltage to top electrode forms the filament and a negative voltage to top electrode attracts the V_O and ruptures the filament (**Fig. 1a**).

In our bulk RRAM devices, we deposited an amorphous, porous, and V_O -rich thick TiO_x layer instead of having a thick crystalline ALD TiO_2 layer. Due to the absence of fast diffusion paths or accumulation sites for V_O defects in the amorphous phase, the V_O defects are not clustered in specific locations that facilitate filament formation. The filament formation is effectively suppressed in an amorphous layer as compared to the crystalline phase in RRAM devices.² V_O defects will drift homogeneously throughout the entire area of the layer rather than forming defects-clustered filaments, following the direction of the electric field, enabling bulk switching instead of filamentary switching. When a positive voltage is applied to the top electrode, the V_O are pushed downwards following the electric field and the V_O concentration

in the TiO_x layer is reduced (**Fig. 1a**). Space-charge-limited-conduction (SCLC) dominates the conduction in the TiO_x layer. Since the V_O concentration in the TiO_x layer is reduced, the SCLC current decreases confirming that the device is reset to a higher resistance state. We successfully confirmed this model by fitting IV characteristics at different resistance states to our model including tunneling and SCLC conduction (**Fig. 3g**) and extracted the trap density (**Fig. 3h**). The trap density is decreased after the device is programmed to higher resistance states using positive bias.

We had added explanations below in the manuscript in the previous round of revisions:

“Bulk switching can be better understood by reviewing filamentary RRAM first. In the filamentary RRAM, V_O defects are well known to be mobile with external electrical and thermal stimuli.¹ Filamentary RRAM needs an initial high voltage electroforming step which forms the V_O defect filaments between two electrodes. Once the filaments have formed, bipolar switching takes place due to the forming and rupturing of the filaments. Meanwhile, the grain boundaries are well known to be the high diffusivity paths of small ions such as oxygens or hydrogens in crystalline oxides.^{3,4} Especially for our polycrystalline filamentary RRAM devices (S1, S2), the grain boundaries play an important role in charge transport and V_O accumulation and diffusion. Due to these diffusivity paths, filament formation and rupture easily occur in the filamentary RRAM devices.

In our bulk RRAM devices, we deposited an amorphous, porous, and V_O -rich thick TiO_x layer instead of having a crystalline ALD TiO_2 layer. Due to the absence of fast diffusion paths or accumulation sites for V_O defects in the amorphous phase, the V_O defects are not clustered in specific locations that facilitate filament formation. The filament formation is effectively suppressed in an amorphous layer as compared to the crystalline phase in RRAM devices.² V_O defects will drift homogeneously throughout the entire area of the layer rather than forming defects-clustered filaments, following the direction of the electric field, enabling bulk switching instead of filamentary switching. When a positive voltage is applied to the top electrode, the V_O are pushed downwards following the electric field and the V_O concentration in the TiO_x layer is reduced. Space-charge-limited-conduction (SCLC) dominates the

conduction in the TiO_x layer. Since the V_O concentration in the TiO_x layer is reduced, the SCLC current decreases confirming that the device is reset to a higher resistance state. We successfully confirmed this model by fitting IV characteristics at different resistance states to our model including tunneling and SCLC conduction (**Fig. 3g**) and extracted the trap density (**Fig. 3h**). The trap density is decreased after the device is programmed to higher resistance states using positive bias.”

- 1 Waser, R. & Aono, M. Nanoionics-based resistive switching memories. *Nature materials* **6**, 833-840 (2007).
- 2 Grossi, A. *et al.* Electrical characterization and modeling of 1T-1R RRAM arrays with amorphous and poly-crystalline HfO₂. *Solid-State Electronics* **128**, 187-193 (2017).
- 3 Park, J., Yoon, H., Sim, H., Choi, S.-Y. & Son, J. Accelerated hydrogen diffusion and surface exchange by domain boundaries in epitaxial VO₂ thin films. *ACS nano* **14**, 2533-2541 (2020).
- 4 Navickas, E. *et al.* Fast oxygen exchange and diffusion kinetics of grain boundaries in Sr-doped LaMnO₃ thin films. *Physical Chemistry Chemical Physics* **17**, 7659-7669 (2015).